# PROCEEDINGS A

climatology, meteorology, atmospheric science

climate change, climate ethics, uncertainty, atmospheric circulation, climate impacts

**Author for correspondence:**
Theodore G. Shepherd
e-mail: theodore.shepherd@reading.ac.uk

# Storyline approach to the construction of regional climate change information

Theodore G. Shepherd

Department of Meteorology, University of Reading, PO Box 243, Earley Gate, Whiteknights, Reading RG6 6BB, UK

TGS, 0000-0002-6631-9968

Climate science seeks to make statements of confidence about what has happened, and what will happen (conditional on scenario). The approach is effective for the global, thermodynamic aspects of climate change, but is ineffective when it comes to aspects of climate change related to atmospheric circulation, which are highly uncertain. Yet, atmospheric circulation strongly mediates climate impacts at the regional scale. In this way, the confidence framework, which focuses on avoiding type 1 errors (false alarms), raises the prospect of committing type 2 errors (missed warnings). This has ethical implications. At the regional scale, however, where information on climate change has to be combined with many other factors affecting vulnerability and exposure—most of which are highly uncertain—the societally relevant question is not 'What will happen?' but rather 'What is the impact of particular actions under an uncertain regional climate change?' This reframing of the question can cut the Gordian knot of regional climate change information, provided one distinguishes between epistemic and aleatoric uncertainties—something that is generally not done in climate projections. It is argued that the storyline approach to climate change—the identification of physically self-consistent, plausible pathways—has the potential to accomplish precisely this.

## 1. Introduction

Although there is high confidence in thermodynamic aspects of climate change (global warming, sea-level

change in precipitation

**Figure 1.** Projected changes in precipitation (in %) over the twenty-first century under a high climate forcing scenario (RCP8.5). Stippling indicates where the multi-model mean change is large compared with natural internal variability in 20 year means (greater than two standard deviations) and where at least 90% of models agree on the sign of change. Hatching indicates where the multi-model mean change is small compared with internal variability (less than one standard deviation), but this does not mean that individual model changes are small. From the Summary for Policymakers of [2]. (Online version in colour.)

rise, atmospheric moistening, melting of ice), the levels of confidence concerning dynamical aspects of climate change, such as the location and strength of storm tracks, are much lower [1]. None of the three key lines of evidence used in climate change science—predicated by accepted theory, detected in observations and consistently represented in climate models—applies to aspects of climate change that are closely related to large-scale atmospheric circulation. This includes, notably, mean precipitation changes over many of the most populated regions on Earth (figure 1). It is in striking contrast to thermodynamic aspects of change, at least when sufficiently aggregated [3], where all three lines of evidence apply [2].

Lack of agreed upon theoretical predictions is related to the fact that different drivers of change can act in opposite directions, so the result is often a small difference of large terms [4,5]. Lack of detection in observations is related to the small signal-to-noise ratio of forced circulation changes, reflecting the fact that climate variability is primarily a dynamical phenomenon [6]. Lack of model agreement is related to both these issues, and to the fact that circulation changes are often quite sensitive to model biases, which can be substantial [7–9].

Furthermore, thermodynamic aspects of climate can be described by extensive quantities (e.g. heat content or ocean volume), which can be readily aggregated, and strong conclusions can be drawn from thermodynamic principles alone, often in terms of global budgets [10]. By contrast, circulation aspects of climate are inherently regional, and involve dynamics (Newton's second law) as well as thermodynamics. Since dynamics is also inherently chaotic, the challenge of atmospheric circulation should come as no surprise.

Ways must therefore be found to construct useful scientific information on the regional scale, and even on the local scale, that reflect an appropriate level of uncertainty yet retain the relevant information about climate risk. It has recently been argued [11] that *storylines*—physically self-consistent unfoldings of past events, or of plausible future events or pathways—provide a potential way forward, both for the interpretation of the observed record and for the description of plausible futures. However, storylines are inherently subjective and thus would seem to be at odds with more probabilistic approaches, which give the appearance of objectivity. The purpose of this paper is to place storylines within a broader epistemological framework.

It is first shown (§2) how the standard, confidence-based framework for the construction of climate information prioritizes reliability (the avoidance of type 1 errors, or false alarms) over informativeness (the avoidance of type 2 errors, or missed warnings), and thus has ethical implications. It follows that there is no such thing as value-free climate science. In §3, the

difference between epistemic and aleatoric (random) uncertainty is shown to be critical to the treatment of climate risk. Since epistemic uncertainty is deterministic and inherently subjective, it follows that there is no objective basis for a probabilistic approach, and no such thing as objective climate information. This motivates a reframing of the climate risk question from the ostensibly objective prediction space into the explicitly subjective decision space (§4). Finally, it is shown in §5 how such a reframing can be cast within the mathematical framework of a causal network, thereby reconciling storyline and probabilistic approaches.

## 2. The confidence straightjacket

The most authoritative statements on physical aspects of climate change come from Working Group I (WGI) of the Intergovernmental Panel on Climate Change (IPCC). In the Summary for Policymakers of the last (Fifth) IPCC WGI Assessment Report [2], atmospheric circulation is scarcely mentioned, and all the statements of confidence are based on thermodynamics. This remarkable fact evidences better than anything else the lack of scientific consensus on dynamical aspects of climate change. Moreover, the statements of confidence are crafted to be reliable, generally by emphasizing global rather than regional aspects of change. A good example is the headline statement on the water cycle:

> Changes in the global water cycle in response to the warming over the 21st century will not be uniform. The contrast in precipitation between wet and dry regions and between wet and dry seasons will increase, although there may be regional exceptions. [2, p. 20]

This statement is based on the sound physical principle that, all else being equal, a moister atmosphere will exhibit an accelerated hydrological cycle [10]. The statement achieves its reliability in the tropics by including oceanic regions (figure 1); indeed, a key observation supporting the statement is the increased salinity in the subtropical upper oceans (due to increased evaporation). However, it is precipitation over land that matters for climate impacts, and there have been many studies showing that the 'wet get wetter, dry get drier' paradigm does not hold over land regions [12–14], as is reflected in the general lack of stippling over these regions (apart from the high northern latitudes) in figure 1. The statement is perfectly reliable as an explanation of how the global climate system works, but it does not provide useful information at the regional scale, as the final caveat makes clear. In this way, reliability is achieved at the price of informativeness.

To find a high-level statement on dynamical aspects of climate change in the IPCC WGI Fifth Assessment Report, one must look one level down, in the Technical Summary [2]. The statements are uniformly characterized by low levels of confidence and a lack of informativeness at the regional scale. An illustrative example is the statement on changes in Northern Hemisphere (NH) storm tracks, which are an important determinant of mid-latitude weather:

> Substantial uncertainty and thus low confidence remains in projecting changes in NH storm tracks, especially for the North Atlantic basin. [2, p. 90]

Furthermore, IPCC WGI uses a likelihood scale in which the term 'unlikely' is used to describe likelihoods of up to 33%. This terminology seems rather perverse from a lay perspective; in most areas of life, one would pay attention to likelihoods that high, especially if the consequences were serious—as they are with climate change. (Would you board an aeroplane if you were told that it had a 33% chance of crashing?) Yet in the WGI report, the term 'unlikely' is generally used to dismiss rather than to highlight a possibility. Consider this example from the Technical Summary, again with reference to the North Atlantic storm track:

> . . . it is unlikely that the response of the North Atlantic storm track is a simple poleward shift [2, p. 108]

The context here is that, despite the lack of an agreed upon theoretical explanation, the concept of a poleward storm track shift under climate change has become a general expectation [5]. However, projected changes in the North Atlantic storm track do not conform to that expectation [15]. An equivalent version of the statement would be ' . . . it is likely that the response of the North Atlantic storm track [to climate change] is not a simple poleward shift'. Because, in the present state of knowledge, a consensus statement could not be crafted on what was likely to happen, the authors instead chose to emphasize what was not likely to happen. Yet, there are several possibilities for what might happen, each with its own implications for climate risk, which could have been articulated (e.g. [16]). However, the simultaneous consideration of contradictory futures is not naturally expressed through statements of confidence. Thus, reliability is again achieved at the price of informativeness.

These examples illustrate the fact that by employing a confidence framework, which seeks to attribute what has happened and to predict what will happen (for a given climate forcing scenario), climate science winds up in something of a straightjacket when it comes to aspects of regional climate change that are closely related to large-scale atmospheric circulation, such as drought and storminess.

It is notable in this respect that IPCC Working Group II, which deals with impacts and adaptation, defines climate change as any observed change, not necessarily one that has been attributed to anthropogenic forcing [17]. This is done to avoid the confidence straightjacket, but it creates a knowledge gap between the WGI and WGII science domains [18].

There is always a trade-off to be made between reliability and informativeness [19]. Yet, a focus on reliability, guarding preferentially against type 1 errors (false positives, i.e. false alarms), increases the likelihood of type 2 errors (false negatives, i.e. missed warnings). It follows that much as though climate science might strive to be value free, it cannot be: the way in which climate information is constructed has ethical implications [20]. Lloyd & Oreskes [20] raise the important question of why, in climate science, it has become normative that scientific rigour is associated with a focus on reliability. They point out that the decision on whether to preferentially guard against type 1 or type 2 errors is not a scientific one, but one of values. For example, in deciding whether to bring a new drug to market, one assesses both the drug's efficacy (guarding against type 1 errors) and whether it has any unwanted side effects (guarding against type 2 errors). Similarly, in deciding whether to issue an evacuation order for a city in the face of a forecasted storm, a balance of concern between type 1 and type 2 errors will be considered. Thus, there is nothing unscientific about seeking to guard against type 2 errors.

It would seem entirely appropriate to preferentially guard against type 1 errors when making high-level definitive statements concerning global climate change such as 'Warming of the climate system is unequivocal' [2, p. 4]. However, the framework is not so evidently appropriate when it comes to regional aspects of change (see also [21]). This situation seems to be an example of Kuhn's [22, p. 37] important observation that 'a paradigm can . . . insulate the [scientific] community from those socially important problems that . . . cannot be stated in terms of the conceptual and instrumental tools the paradigm supplies'. Thus, it is imperative to find alternative paradigms.

## 3. Epistemic versus aleatoric uncertainty

Broadly, uncertainty in climate projections arises from three sources: uncertainty in future climate forcing, in the climate system response to that forcing (i.e. the change in climate) and in the actual realization of climate for a particular time window, which is subject to internal variability. The nature of these uncertainties is very different (e.g. [23]). The first depends primarily on human actions and is called the scenario, and the projections are normally made conditional on the scenario. The second is what is known as an *epistemic* uncertainty; there is only one truth, but we do not know what it is. The third is what is known as an *aleatoric* uncertainty; there is a random element to what will occur, whose probability is known to some extent. Any discussion of climate risk must address the central fact that the nature of the second and

third uncertainties is fundamentally different. This is especially important for circulation-related aspects of climate change at the regional scale, for which these two sources of uncertainty tend to dominate the overall uncertainty (see [24] for regional precipitation changes). Yet, it is standard practice in climate science to mingle the two sources of uncertainty together, e.g. in the multi-model ensembles (with one realization taken from each model) that are in such widespread use [2]. In such ensembles, the differences between the individual model projections include both the systematic differences between different model climates (epistemic) and the random differences that arise from the limited sampling of internal variability (aleatoric). Since only the latter possess an underlying probability distribution, this poses challenges in interpretation [25].

We first discuss the uncertainty arising from internal variability, since it is conceptually much easier to deal with. Internal variability is a property of the physical climate system, whose random character arises from the chaotic nature of atmospheric and oceanic dynamics, and which can be characterized from observations. Indeed, the definition of climate includes internal variability, which is characterized through statistical measures such as variances and covariances of physical fields, as well as higher order moments such as skewness or extremes, and includes coherent modes of variability such as the El Niño/Southern Oscillation phenomenon. The uncertainty from internal variability is fundamentally irreducible (leaving aside the possibility of finite-time prediction from specified initial conditions), and users of climate information need to understand that the mantra of 'reducing uncertainty' is inappropriate in this case; rather, the scientific goal is to better quantify the uncertainty. The magnitude of the uncertainty for any particular quantity can be reduced by taking coarser spatial and temporal averages, but that operation changes and may simultaneously reduce the value of the information provided.

The concept of internal variability is not without ambiguity since climate has various sources of non-stationarity, and what is meant by internal variability is conditional on any non-stationary influence, including climate change itself. Furthermore, knowledge of internal variability is limited by the finite observational record, and there is uncertainty in how internal variability will respond to global warming. Nevertheless, in most cases, the main uncertainty in what climate conditions will be experienced at a particular place and time arising from internal variability can be considered to be aleatoric, and thus amenable to a straightforward (i.e. frequentist) probabilistic interpretation. The reliability of model simulations of internal variability can be similarly assessed, at least in principle.

The uncertainty in the climate response to forcing is conceptually very different. It is not a property of the physical climate system; rather, it is a property of a state of knowledge, or degree of belief, and it *can* be reduced as knowledge improves. In contrast with aleatoric uncertainty, which is objective, such epistemic uncertainty is *subjective* [26]. Therefore, treating epistemic uncertainty as if it were aleatoric, with a focus on the multi-model mean as a best estimate, has no epistemological justification. This has been recognized for some time [21,27,28], but the practice continues to be normative (e.g. as in figure 1). It is interesting to consider why this is so, since, in most areas of science, the essential distinction between systematic and random sources of uncertainty is well recognized. One of the reasons may be that the extent of the epistemic uncertainty is not particularly well known. First, climate models are imperfect representations of reality and share many deficiencies, thus they may exhibit a collective bias and fail to explore important aspects of climate change. Second, even within the world represented by climate models, the forced circulation response of any particular model is obscured by internal variability.

As an example of the latter, Deser *et al*. [29] estimate that, for NH wintertime mid-latitude surface pressure (whose spatial gradient provides an indicator of circulation changes), ensemble sizes of around 30 are generally needed to determine the forced decadal changes of a given model over a 45 year period. This is in striking contrast to surface temperature changes, where the signal-to-noise ratio of the forced response is much larger, and even single simulations can be informative. One might be tempted to think that if such a large ensemble size is needed to detect the signal, then the signal must be small. However, Deser *et al*. [29] show that such a change in surface pressure patterns can alter the risk of regional drought or heavy precipitation by a factor of two, which is hardly negligible. Most climate model simulations are performed with much

smaller ensemble sizes, although there is a growing interest in large single-model ensembles in order to better characterize the epistemic uncertainty within current models.

Another conceptual challenge in dealing with the epistemic uncertainty of climate change is that the concept of 'error' is not well defined. Although in principle there may be one truth, it is not knowable: there will never be sufficient observations to define all relevant aspects of future climate; future climate will in any case be non-stationary; and model projections are based on climate forcing scenarios that will not be the ones actually realized. Thus, there has been interest in trying to understand the relationship between model errors in observable aspects of climate and the forced response simulated by that model—so-called emergent constraints (e.g. [30]). Such an approach permits a Bayesian probabilistic interpretation of epistemic uncertainty [31]. However, there is a danger that any such relationship is merely statistical and not causal, and many published emergent constraints have been subsequently debunked (see [32–34]). In any case, subjective choices are required in the application of any such constraints.

That an aleatoric interpretation of multi-model ensembles can blur the climate information contained within those ensembles is not difficult to appreciate. Circulation aspects of climate are related to features such as jet streams. Over Europe during wintertime, some models show an increase in jet strength under climate change and others a decrease (see fig. 4 of [1]); moreover, the location of the changes varies between models. While all models predict a significant jet response somewhere, averaging over the models will lead to a washed-out response. Thus, the multi-model mean may not only be unlikely, but even implausible. The situation is analogous to the idealized case of a bi-modal probability density function, whose mean may not be a physically realizable state.

A related issue is apparent in figure 1. Because precipitation increases in some regions and decreases in others, the multi-model mean change inevitably passes through zero, and will be small compared with internal variability on either side of that line. However, that does not mean that the change in those regions can be expected to be small compared with internal variability; it just reflects uncertainty in the sign of the change. When there are equally plausible futures that point in different directions, averaging those futures buries relevant information and underestimates risk.

The essential point is that epistemic uncertainties are deterministic, which means that they introduce correlations; unless those correlations are accounted for, inferences may be flawed. For example, Madsen *et al*. [35] show that the spread across Coupled Model Intercomparison Project Phase 5 (CMIP5) model projections in temperature and precipitation changes at the gridpoint scale is significantly exaggerated when treating the gridpoints independently, as compared with when the models are ranked by the global-mean changes (where the spread comes mainly from climate sensitivity). This illustrates the general point that, with an inhomogeneous distribution of estimators, one should examine the distribution of responses to a perturbation rather than the overall response of the distribution to the perturbation.

## 4. Reframing the question

If the construction of regional climate information inevitably involves ethical choices, then those choices should be made by the users of the climate information, based on their values. If the uncertainties in the climate information involve a significant epistemic component, then subjectivity is inevitable and the epistemic uncertainties similarly need to be understandable and assessable by the users of the climate information, within their particular context. Both imperatives move the climate risk problem outside the domain of pure climate science. Moreover, the recognition that epistemic uncertainties are deterministic removes the impulse to provide probabilities, which can give the illusion of objectivity and thereby reduce transparency. Instead, epistemic uncertainty can be represented through a discrete set of (multiple) storylines—physically self-consistent, plausible pathways, with no probability attached [11,36].

Rather than asking what will happen (as in the traditional, scenario-driven approach), which we may not be able to answer with any confidence, storylines allow us to ask what would

be the effect of particular interventions—e.g. different climate forcing scenarios, or different adaptation measures—across a range of plausible futures. The latter questions are in any case the societally relevant ones. This reframing of the climate risk question from the prediction space to the decision space avoids the confidence straightjacket. Storylines have much in common with scenario planning and other methods of robust decision-making under uncertainty [37,38]. What is novel is their application to physical climate science, where, perhaps because the system obeys known physical laws, the operative paradigm up to now has been probabilistic, which gives the impression of objectivity.

The different uncertainties that are relevant to climate risk, and the different human decision points, can be broadly represented as follows. There is uncertainty in the future climate forcing, which is mainly anthropogenic in origin, and represents the mitigation options. This combines with the epistemic uncertainty in climate sensitivity to determine the global-mean warming level. While there is some aleatoric uncertainty in the global-mean warming, it is small compared with the forced response on decadal or longer time scales. A given global-mean warming level will be associated with distinct patterns of regional warming (e.g. land warms more than ocean, the Arctic warms more than lower latitudes during the winter season), including changes in lapse rate [39]. These regional warming patterns are largely explainable from thermodynamic principles and thus are fairly well understood, though they have substantial quantitative epistemic uncertainty (including the possibility of tipping points). A given global-mean warming level will also be associated with particular dynamical conditions in any specific region (including the circulation effects of coupled atmosphere–ocean variability), which have a very large aleatoric component but whose forced changes are also highly uncertain. The regional warming patterns and dynamical conditions together produce hazards such as weather or climate extremes, which then combine with the non-climatic anthropogenic factors of vulnerability and exposure to create climate impacts.

This representation of the climate risk problem provides a natural framework for storyline approaches. For example, from the perspective of the Paris Agreement, one may ask the question of what the climate impacts would be at different levels of global-mean warming, and what different mitigation pathways would lead to those warming levels [40]. The epistemic uncertainty in climate sensitivity now no longer affects the estimation of climate impacts, but is instead relevant to the carbon budget allowed by the given level of warming. The epistemic uncertainty in future dynamical conditions (for a given level of global-mean warming) can then be managed via storylines, the simplest of which is that the changes in hazard are dominated by the thermodynamic effects arising from the regional temperature changes, with the forced changes in dynamical conditions assumed to be negligible. Given the large uncertainties in the forced dynamical changes, this can be considered a reasonable null hypothesis for climate change [41,42], and it is far from uninformative. It is in fact the basis for all of the predicted changes in extremes shown in table SPM.1 of the IPCC WGI Fifth Assessment Report [2]. It also underlies the 'surrogate climate change' (also known as 'pseudo-global warming') methodology [39,43], which is widely used in regional climate change simulations, and the circulation-analogue methodology [44], which is widely used in extreme-event attribution. However, specific storylines of forced circulation change can also be considered [16,42].

Reframing the climate risk question in this way increases the signal-to-noise ratio of the climate information by explicitly accounting for the correlated nature of epistemic uncertainty. An example is provided in figure 2. The Mediterranean region receives most of its precipitation during the winter season, so the predicted wintertime drying of the region, which is a robust feature of climate model projections (figure 1), has important consequences. The extent of the drying will depend on the global warming level, and it is relevant to ask, for instance, what would be the difference between 1.5°C and 2.0°C of global warming. However, the extent of the drying will also depend on the pattern of circulation change in the region—an epistemic uncertainty—which can be characterized by physically coherent storylines [16]. Considering just the range between the low-impact and high-impact storylines shown in figure 2, the difference in drying between 1.5°C and 2.0°C of global-mean warming under the standard probabilistic

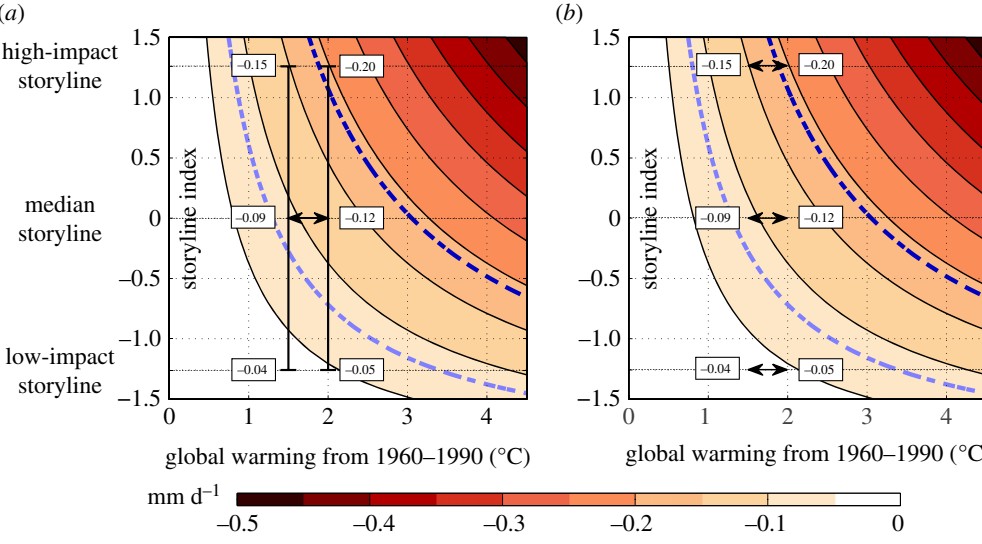

**Figure 2.** Projected average wintertime precipitation change (in mm d$^{-1}$) over the Mediterranean basin plotted as a function of global warming level (in °C) relative to 1960–1990 and a 'storyline index' that represents the uncertainty in the pattern of circulation change in the region. The high-impact storyline corresponds to the combination of strong tropical upper tropospheric amplification of surface warming and a strengthening of the stratospheric polar vortex, and the low-impact storyline to weak tropical upper tropospheric amplification of surface warming and a weakening of the polar vortex. The light blue dashed line represents a magnitude of change that is statistically detectable, and the dark blue dashed line corresponds to one standard deviation of the interannual variability. (a) The standard representation of the difference between global warming levels of 1.5°C and 2.0°C is shown, taking the low- and high-impact storylines as spanning a range of uncertainty. (b) Differences are shown conditioned on different storylines. Adapted from [16]. (Online version in colour.)

framing is the difference between 0.09 [0.04 to 0.15] and 0.12 [0.05 to 0.20] mm d$^{-1}$ (figure 2a), which would be considered indeterminate within the stated uncertainties. The storyline framing of the difference is, by contrast, a deterministic 0.04 versus 0.05 for the low-impact circulation storyline, 0.09 versus 0.12 for the median storyline and 0.15 versus 0.20 for the high-impact storyline (figure 2b). This is a more informative way of representing the uncertainty, because it quantifies different plausible outcomes. For reference, 0.08 mm d$^{-1}$ corresponds to a change that is statistically detectable, and 0.19 to one standard deviation of the interannual variability—quite a large change, probably requiring significant adaptation measures. The distinction between the two approaches is analogous to that between accuracy and precision; sometimes, the latter is all that is needed for decision-making.

Storylines are ideal vehicles for quantifying the impacts of climate change and adaptation measures. They provide a way of dealing with singular historical events, which within the probabilistic framework are merely accidents within a phase space of unrealized possibilities, yet often provide benchmarks for resilience; and with the local context, where the human element becomes part of the analysis rather than a confounding factor. For example, rather than seeking to determine the recurrence likelihood of a particularly damaging storm (an inherently fuzzy question since every storm is unique), one can ask how much worse the flooding would be in a warmer, moister climate [41], or under a particular urban development scenario. Such conditioning of the question enormously reduces the dimension of the problem and thereby allows the use of much more realistic modelling tools, which users of climate information can relate to. In this way, the storyline approach addresses the needed reframing of the climate risk problem while representing the epistemic uncertainties in a traceable manner.

That there is relevant information concerning climate risk contained even in single historical events is illustrated in figure 3, which shows a small region in central France during one day in

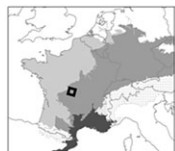

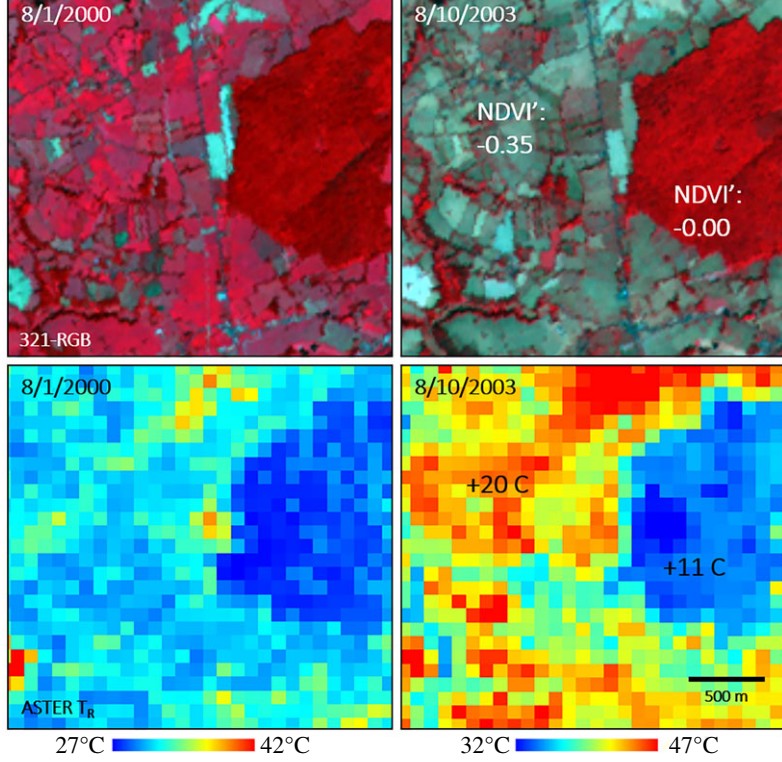

**Figure 3.** Surface conditions derived from infrared remote sensing for a small region in central France, for 1 August 2000 (left panels) and 10 August 2003 (right panels). The top panels show the normalized difference vegetation index (NDVI), with the red colours indicative of vegetation. The lower panels show the radiometric temperature, with the colour scale at the bottom. The distance scale is shown in the lower right panel, and the values given in the right panels indicate the average differences in those parts of the scene between the left and right panels. Adapted from [45]. (Online version in colour.)

August 2000 and another day in August 2003 during the severe heat wave that affected Europe that summer [45]. From a statistical perspective, it may seem meaningless to compare two single days because they will each be strongly influenced by synoptic variability. However, the images show that the crops and grasses in the agricultural plots died out during the 2003 heat wave, and the surface temperature difference between the 2 days over those parts of the scene was 20°C, versus only 11°C in the forested region. Since a difference of 9°C over a distance of several hundred metres cannot be explained by synoptic variability (which has much larger correlation scales), this clearly shows the impact of land cover on the climate risk from heat waves. (Moreover, the average temperature difference in the agricultural plots rises to 24°C if the hedgerows are excluded, and the temperature difference in fields that were bare in both 2000 and 2003 is 11°C.) While it may not be possible to predict the future statistics of heat waves in this region, it is possible to make informative statements about how those heat waves would be affected by land cover and thus inform adaptation strategies.

The tension between global and local descriptions (in time or space) is not unique to climate science, of course. It arises in any scientific context where statistical power is achieved by aggregation over an inhomogeneous population, and thus blurs information. There is a growing move in many fields towards analysis methods that aim to consider information in context rather than in aggregate, especially when that information is sparse (e.g. safety in healthcare [46]). Storyline approaches to climate risk can be seen as part of that movement.

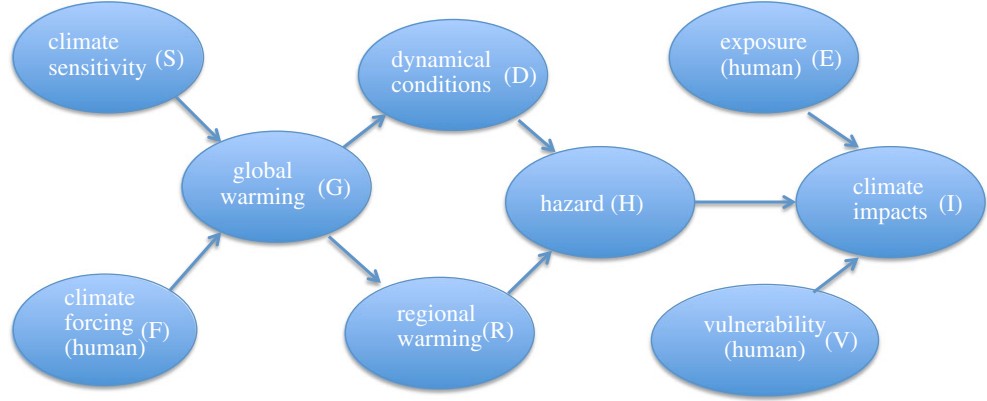

**Figure 4.** A causal network describing regional climate risk. The arrows indicate the directions of causal influence. See text for details. (Online version in colour.)

## 5. Causal networks

In the above, storylines have been presented in contrast with probabilistic representations of uncertainty. However, if storylines are to provide an alternative scientific paradigm for the construction of regional climate change information, they must be somehow reconcilable with the conventional, probabilistic approach, in order to effectively bridge between climate science and climate impacts, and from the global to the local scale.

The narrative description of the regional climate risk problem in the previous section is represented graphically in figure 4. Figure 4 is a *directed acyclic graph*, which means that the climate risk problem can be represented mathematically as a causal network [47,48]. This observation provides the key to reconciling storyline and probabilistic approaches. Following [48], a joint probability of $n$ variables $P(x_1, \ldots, x_n)$ can be expressed as the product of conditional probabilities $P(x_j \mid pa_j)$, where $pa_j$ are the 'parent' factors influencing $x_j$, according to

$$P(x_1, \ldots, x_n) = \prod_j P(x_j|pa_j). \tag{5.1}$$

The representation (5.1) factorizes the uncertainty, which is extremely useful when the different uncertainties have rather different characteristics, as in the climate risk problem. A storyline $x_i = x_i'$ for a particular $i$ can be defined by imposing that particular condition within (5.1), represented symbolically by $\hat{x}_i'$, which leads to [48, pp. 72–73]

$$P(x_1, \ldots, x_n|\hat{x}_i') = \begin{cases} \prod_{j \neq i} P(x_j|pa_j) = \dfrac{P(x_1, \ldots, x_n)}{P(x'_i|pa_i)} & \text{if } x_i = x'_i \\ 0 & \text{if } x_i \neq x'_i. \end{cases} \tag{5.2}$$

The expression (5.2) is thus a *truncated factorization* of the expression (5.1) for the unconditional probability, representing a blend of probabilistic and deterministic factors. Multivariate storylines can be treated by repeated application of this procedure. In this way, storylines can be cast within the context of a probabilistic framework.

We illustrate this for the system represented in figure 4. The traditional scenario-driven prediction problem aims to estimate the joint probability of the climate state conditional only on the climate forcing $F$

$$P(H, D, R, G, S|F). \tag{5.3}$$

According to the causal linkages represented in figure 4, this factorizes to

$$P(H|D, R)P(D|G)P(R|G)P(G|S, F)P(S). \tag{5.4}$$

Within this perspective, it is necessary to have knowledge of the climate sensitivity $S$. However, from the perspective of the Paris Agreement, one can define a storyline consisting of a particular global warming level, say $G = G_1$, which specifies $G$ deterministically. This condition blocks the influence of $S$, leaving the truncated factorization

$$P(H|D, R)P(D|G = G_1)P(R|G = G_1), \tag{5.5}$$

where now the hazard $H$ depends only on the dynamical conditions $D$ and the regional warming $R$. Note that (5.5) does not imply that $D$ and $R$ are independent; they share a common dependence through $G$, hence storylines of $R$ may be correlated with storylines of $D$. This is precisely the basis of the approach of [16].

Interestingly, imposing a global-mean warming target builds in a relationship between the climate sensitivity $S$ and the climate forcing $F$. This is in contrast with the traditional scenario-driven formulation of climate risk, where these quantities are treated as independent. [In (5.3), $S$, as a property of the climate system, would be assumed independent of $F$.] Such a relationship expresses the policy-relevant information that society will need to act more aggressively on controlling emissions if climate sensitivity turns out to be high, but may allow itself more time if climate sensitivity turns out to be low.

If $R$ is taken to be a deterministic function of $G$, i.e. the uncertainty in $R$ is considered to be mainly epistemic, then (5.5) simplifies to

$$P(H|D, R = R_1)P(D|G = G_1), \tag{5.6}$$

where $R_1 = R\ (G_1)$. The first term in (5.6) represents the thermodynamic effects of a particular regional warming $R_1$ on $H$, given knowledge of $D$, while the second term represents the dynamical effects of climate change. As already discussed, the epistemic uncertainty in the latter can be very high, but is representable through storylines. The simplest storyline is that the dynamics remains unchanged, in which case the conditionality in the second term drops out and we are left with

$$P(H|D, R = R_1)P(D), \tag{5.7}$$

where $P(D)$ can be based, for example, on observations. This is exactly the formulation of the 'surrogate climate change' methodology mentioned earlier, which is widely used in regional climate change simulations. However, one can certainly also specify different dynamical storylines to represent plausible changes in dynamics. Since (5.6) essentially describes the regional climate modelling paradigm, it may provide a useful framework for the construction of regional climate change information and the design of ensembles of simulations using regional climate models, including the representation of particularly extreme forms of internal variability.

Without this factorization of the probabilities, the regional climate risk problem for a given global warming level is representable instead in the form

$$P(H, D, R|G), \tag{5.8}$$

which lends itself to a probabilistic interpretation of the dynamical aspects of climate change. This hides the implicit assumptions concerning the epistemic uncertainties that are made explicit in the representation (5.6). Moreover, the comparatively unconditional nature of (5.8) requires the use of global models, whereas (5.6) permits the use of regional models, which can provide a more physically realistic representation of regional climate risk [49–51].

By casting storylines within the context of a probabilistic framework, it becomes clear that there is nothing to prevent assigning probabilities to storylines, if the scientific basis exists to do that. At the very least, physically implausible behaviours could be excluded [50]. As epistemic uncertainties are reduced, this knowledge can be immediately incorporated into a revised risk analysis. Thus, storylines provide a very flexible, transparent representation of epistemic uncertainty.

Not only do causal networks reconcile storyline and probabilistic approaches to climate risk, they are also ideally suited for moving the risk question into the decision space. That is because

the calculus of causal networks explicitly allows the consideration of counter-factual outcomes [48], and decision-making is precisely the consideration of counter-factual outcomes. Within this context, storylines correspond to what Halpern & Pearl [52, p. 889] define as explanations: 'a fact that is not known for certain but, if found to be true, would constitute an actual cause of the fact to be explained, regardless of the agent's initial uncertainty'.

More generally, causal networks are a way of combining expert knowledge with probability [47]. The factorization (5.1) allows for the ready incorporation of knowledge within a local semantics, and yields results that are comprehensible to humans [53]. In the published Discussion of Lauritzen & Spiegelhalter [47, p. 210], Pearl invokes the following statement (attributed to G. Halter): 'Probability is not really about numbers; it is about the structure of reasoning'. Making the subjective assumptions explicit leads to transparency in the subsequent analysis [54] and provides an audit trail for decision-makers [55]. This is important since, as Beven [55, p. 1661] puts it, 'Decision and policy makers are . . . far more interested in evidence than uncertainty'.

The challenge for regional climate change science then becomes that of constructing suitable causal networks. Causal networks are necessarily a simplification, because they entail the reduction of continuous fields to a finite-dimensional system. However, they very much correspond to how climate scientists reason. For example, the El Niño variability in tropical sea-surface temperatures drives a Rossby-wave teleconnection pathway which affects circulation and weather regimes in the mid-latitudes, and all these elements can be represented to a reasonable extent with physical climate indices. Thus, atmospheric dynamics already provides the building blocks for the construction of causal networks relevant to regional climate risk. (In practice, the 'dynamical conditions' node in figure 4 could be expanded into a subnetwork.) Comprehensive climate simulation models are still needed to explore uncertainty space, but causal networks can provide the diagnostic framework within which to extract the relevant climate information from those simulations, and combine it with other sources of information in a format that is suitable for decision-making.

The causal network depicted in figure 4 incorporates two emergent aspects of climate change. Both aspects are simplifications, but they are extremely powerful and are widely used in the interpretation of climate information. The first is what is known as 'pattern scaling' [56,57]: namely that regional climate change is a function of global-mean warming. In practice, the patterns of regional warming are time dependent [58] so are different for transient and equilibrated warming levels, and short-lived climate forcers such as aerosol can have distinct regional effects [59]. Such additional degrees of freedom, as well as global tipping points, could be incorporated by making the node $G$ suitably multivariate. The second emergent aspect is the distinction between thermodynamic and dynamical aspects of regional climate change, which has already been discussed. While the distinction is not precise and has its limitations, it is useful (e.g. [60]); it has even been used for the last two Dutch climate change scenarios [61]. As with the other simplifications implicit in figure 4, e.g. the lack of any arrows pointing back from the right to the left, the validity of all these simplifications can be assessed *a posteriori*.

Note that linearity is not assumed in causal networks. However, if certain relationships can be shown to be linear to a suitable level of approximation for the problem at hand, then the analysis is enormously simplified. This is generally necessary for any observational analysis, because of the limited sample size [62].

## 6. Discussion

This paper has argued that the storyline approach to regional climate change information avoids the straightjacket that hampers the standard confidence-based approach, by allowing a reframing of the climate risk question from the prediction space into the decision space. While in principle such a reframing is possible from probabilistic estimates of risk, the challenge for regional climate change information is that the level of epistemic uncertainty is sufficiently high that subjective choices must inevitably be made, and the range of users sufficiently inhomogeneous that there is no consensus on values. Under such conditions, probabilistic 'rational-choice' approaches to

decision-making are ineffective [63,64] and the decision framework needs to be one where the subjective and ethical choices are both flexible and transparent [65]. Since epistemic uncertainty is inherently deterministic and subjective, there is no imperative to represent it probabilistically [23], and probabilistic representations can give a false impression of objectivity.

The reframing of the risk question from the prediction space to the decision space may seem uncomfortable from a physical science perspective, but is in fact quite orthodox from the perspective of statistical inference. Despite the widespread use of $p$-values as an ostensibly objective measure of statistical significance, the inference derived from data concerning a particular hypothesis is far from a straightforward matter and involves many assumptions [66]. In the Neyman–Pearson framework, the inference problem is regularized by placing it in a decision context between two alternative hypotheses, which takes into account the possibility of both type 1 and type 2 errors [67]. In the Bayesian framework, the strength of evidence between these alternative hypotheses ($H_1$ and $H_2$) provided by the data $D$ is given by

$$\frac{P(H_2|D)}{P(H_1|D)} = \frac{P(D|H_2)}{P(D|H_1)} \frac{P(H_2)}{P(H_1)}, \tag{6.1}$$

which follows directly from Bayes' theorem. The Bayes factor $P(D|H_2)/P(D|H_1)$ is independent of the prior likelihoods $P(H_2)$ and $P(H_1)$, so can be considered objective, but it does not represent any sort of absolute knowledge—only an increment in knowledge, relative to the prior beliefs.

Moving the climate risk problem out of the domain of pure climate science requires humility on the part of climate scientists. To quote Funtowicz & Ravetz [63, pp. 750–751]—who used sea-level rise as an example—'the traditional domination of "hard facts" over "soft values" [is] inverted . . . traditional scientific inputs . . . become "soft" in the context of the "hard" value commitments that will determine the success of policies for mitigating the effects of [climate change]'. Indeed, it has been argued that humility is one of the four core elements—the others being integrity, transparency and collaboration—that should be intrinsic to the production of regional climate information [68]. In this way, the goal is not so much to be authoritative, which has something of a gatekeeper connotation, but to be trustworthy [69]. This involves a loss of control, because one's trustworthiness is a judgement made by others.

This perspective also involves an acknowledgement that climate-relevant decisions, especially at the local scale, are not usually made on the basis of climate change alone but involve many other changing factors, most of which are highly uncertain. If climate impacts $I$ are a product of hazard $H$, vulnerability $V$ and exposure $E$, then, conceptually, the anthropogenic changes in $I$ can be represented as

$$\delta I = \delta(HVE) = HV\delta E + HE\delta V + VE\delta H. \tag{6.2}$$

It may well be that the largest terms on the right-hand side of (6.2) are the first two, where it is the combination of climate and weather *variability* with changing vulnerability and exposure that is the main determinant of climate risk [70]. In this case, the decision framework is not so much that of dealing with climate change as it is that of bringing climate information into decisions that need to be made in any case. There are calls for this sort of complex-systems thinking in other areas of science, such as public health [71, p. 2602]: 'Instead of asking whether an intervention works to fix a problem, researchers should aim to identify if and how it contributes to reshaping a system in favourable ways'.

To return to Kuhn [22], the construction of regional climate change information is not most usefully viewed as a search for an objective truth, but rather as a search for more complete descriptions of the realities that people have experienced and may experience in the future, and how those depend on contingent factors that are under human control. Kuhn's version of the Bayesian perspective described above, and the cutting of the Gordian knot it enables, is as follows [22, p. 170]: 'If we can learn to substitute evolution-from-what-we-know for evolution-toward-what-we-wish-to-know, a number of vexing problems may vanish in the process'. The role of climate scientists is to bring physical knowledge of the climate system into such an enterprise.

Data accessibility. This article has no additional data.

**Competing interests.** I declare I have no competing interests.

**Funding.** This work was supported by the European Research Council (Advanced Grant 339390). The support provided through the Grantham Chair in Climate Science is also gratefully acknowledged.

**Acknowledgements.** Useful comments were received from Michaela Hegglin, Wilco Hazeleger, Lisa Lloyd, Jana Sillmann and the anonymous reviewers.

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
