## [Reviewer comments · Proceedings. Mathematical, Physical, and Engineering Sciences]

Review History

RSPA-2019-0013.R0 (Original submission)

Review form: Referee 1

Is the manuscript an original and important contribution to its field?

Yes

Is the paper of sufficient general interest?

Yes

Is the overall quality of the paper suitable?

Yes

Quality of the paper

An outstanding paper of the highest international importance; a major contribution to the field: must be published.

Can the paper be shortened without overall detriment to the main message?

No

Do you think some of the material would be more appropriate as an electronic appendix?

No

For papers with colour figures – is colour essential?

Yes

If there is supplementary material, is this adequate and clear?

Yes

Are there details of how to obtain materials and data, including any restrictions that may apply?

Yes

Do you have any ethical concerns with this paper?

No

Recommendation?

Accept with minor revision (please list in comments)

Comments to the Author(s)

Please see attached comments.

Review form: Referee 2 (Heini Wernli)

Is the manuscript an original and important contribution to its field?

Yes

Is the paper of sufficient general interest?

Yes

Is the overall quality of the paper suitable?

Yes

Quality of the paper

An outstanding paper of the highest international importance; a major contribution to the field: must be published.

Can the paper be shortened without overall detriment to the main message?

No

Do you think some of the material would be more appropriate as an electronic appendix?

No

For papers with colour figures – is colour essential?

Yes

If there is supplementary material, is this adequate and clear?

Not applicable

Are there details of how to obtain materials and data, including any restrictions that may apply?

Not applicable

Do you have any ethical concerns with this paper?

No

Recommendation?

Accept with minor revision (please list in comments)

Comments to the Author(s)

See attached pdf file.

Review form: Referee 3 (Wilco Hazeleger)

Is the manuscript an original and important contribution to its field?

Yes

Is the paper of sufficient general interest?

Yes

Is the overall quality of the paper suitable?

Yes

Quality of the paper

A good paper worth publishing in Proceedings.

Can the paper be shortened without overall detriment to the main message?

No

Do you think some of the material would be more appropriate as an electronic appendix?

No

For papers with colour figures - is colour essential?

Yes

If there is supplementary material, is this adequate and clear?

No

Are there details of how to obtain materials and data, including any restrictions that may apply?

No

Do you have any ethical concerns with this paper?

No

Recommendation?

Major revision is needed (please make suggestions in comments)

Comments to the Author(s)

The paper is a pleasure to read, insightful and gives a perspective and review of casting decision relevant climate information under (large) uncertainty. A framework is discussed where dominance of epistemic uncertainty leads to an alternative way of obtaining and presenting decision relevant climate information. Storytelling as a means to deal with the subjectivity and complexity of compounded (conditioned or truncated) probabilities is described and a causal network approach is laid out where this is reconciled with traditional frequentist approaches. I find this framework the most interesting part of the paper, but I am biased myself as I worked on storyline approaches.

Given the relevance and importance of the subject and the increasing scientific discussions on this, in particular on the methodology, in the physical climate science community, I would like to see this paper published. The scientific novelty and rigor could be stronger, but as a perspective this is a very relevant and good contribution. It serves the purpose of making physical climate scientists aware how climate information are conveyed and it offers a framework beyond traditional flow of information from global models to impact and decisions. I agree with the author that this traditional approach has very serious problems. The paper is not entirely original though, but it is very nice to see some new aspects addressed. The paper is more of the nature of a perspective paper. I have some reservations which can be addressed by the author:

- One of them is the lack of acknowledgment of existing literature and approaches. The author is likely aware of previous work, but could cite relevant literature more. In particular most elements of the first part the paper on different types of uncertainties have been published in detail before, including the consequences and relevance for decision making for climate (both mitigation and adaptation). The work of Petersen (now at UCL), including his PhD thesis (published as a book as well), goes in great depth on this subject. Also, earlier work of Dessai, Hulme, Lempert and van der Sluijs from the early 2000s onward is scarcely referenced while many of the thoughts put down here can be found in those papers as well. They have been systematically addressed. Again, both for the global mitigation problem as well as the regional climate adaption problems this has been done. The current paper makes the connection to the physical climate science more clear though.

- The author should be more careful with the wording on ethics. I don't think this paper goes into sufficient depth on norms and values to include strong statements on ethics (e.g. the wording in the abstract is rather strong compared to the content of the paper). Although of large relevance, it takes another study to highlight and reason on ethics (and indeed, this has been done as well, but often outside the realm of the physical climate scientists). Also, we should be aware that apparently the traditional approaches are very decision relevant as well, as they have had impact on regional and local adaptation decisions (e.g. spatial planning, infrastructures). To call these approaches almost unethical is debatable.

- The separation of dynamical and thermodynamical processes of climate change can be problematic, as the author acknowledges, as they cannot be separated well. This particular true for tropical and global circulation patterns where convection is important, but for many regional climate adaptation problems in the midlatitudes the separation is mostly useful (but not for summer continental situations). For example, this is clear from the successful Dutch climate scenarios which already separated a regional expression of the global mean temperature change and the regional dynamical changes into 4 possible futures. This was published in 2006 already and in 2014 followed up. In that year these scenarios were complemented by storylines or Tales of current and future weather analogues where the underlying hypothesis was that dynamical changes are harder to predict or project than thermodynamic responses given a dynamical pattern. Indeed the paradigm approach of Kuhn could already be applied to those developments. The current reflection of dynamics - thermodynamic separation therefore needs more embedding

in existing literature and practices.

- A very relevant aspect of the suggested approach is the plausibility of the scenario or storyline. I would like to see more discussion on this. It is only very shortly mentioned at the end of the paper. In particular in relation to the plausibility of dynamical change in global models that drive regional models (with factual, surrogate warming, counterfactual etc boundary conditions).

- Even though this paper is about decision relevance, it takes very much a physical climate scientist point of view only. Most of the discussion and the references to literature is on the physical climate aspect. All the figures are traditional IPCC-like climate variables and metrics and still far away from the practice of decision making in the public or private sector. The land use example is a bit of an exception, but still this is not on decisions, but shows that at regional scales the boundary conditions of the physical climate system conditions the response. I don't see this as a fundamental problem, but it would be good if the author acknowledges that this is the case. There is a wealth of possibilities there, some examples would help.

In summary, I think this is a very useful contribution to the discussion on decision relevant regional climate change information.

Wilco Hazeleger

Decision letter (RSPA-2019-0013.R0)

25-Mar-2019

Dear Professor Shepherd,

On behalf of the Editor, I am pleased to inform you that your Manuscript RSPA-2019-0013 entitled "Storyline approach to the construction of regional climate-change information" has been accepted for publication subject to minor revisions in Proceedings A. Please find the referees' comments below.

The reviewer(s) have recommended publication, but also suggest some minor revisions to your manuscript. Therefore, I invite you to respond to the reviewer(s)' comments and revise your manuscript. Please note that we have a strict upper limit of 28 pages for each paper. Please endeavour to incorporate any revisions while keeping the paper within journal limits. Please note that page charges are made on all papers longer than 20 pages. If you cannot pay these charges you must reduce your paper to 20 pages before submitting your revision. Your paper has been ESTIMATED to be 18 pages. We cannot proceed with typesetting your paper without your agreement to meet page charges in full should the paper exceed 20 pages when typeset. If you have any questions, please do get in touch.

It is a condition of publication that you submit the revised version of your manuscript within 7 days. If you do not think you will be able to meet this date please let me know in advance of the due date.

To revise your manuscript, log into <https://mc.manuscriptcentral.com/prsa> and enter your Author Centre, where you will find your manuscript title listed under "Manuscripts with Decisions." Under "Actions," click on "Create a Revision." Your manuscript number has been appended to denote a revision.

You will be unable to make your revisions on the originally submitted version of the manuscript. Instead, revise your manuscript and upload a new version through your Author Centre.

IMPORTANT: Your original files are available to you when you upload your revised manuscript. Please delete any redundant files before completing the submission process.

In addition to addressing all of the reviewers' and editor's comments, your revised manuscript **MUST** contain the following sections before the reference list (for any heading that does not apply to your work, please include a comment to this effect):

- Acknowledgements
- Funding statement

See <https://royalsociety.org/journals/authors/author-guidelines/> for further details.

When uploading your revised files, please make sure that you include the following as we cannot proceed without these:

- 1) A text file of the manuscript (doc, txt, rtf or tex), including the references, tables (including captions) and figure captions. Please remove any tracked changes from the text before submission. PDF files are not an accepted format for the "Main Document".
- 2) A separate electronic file of each figure (tif, eps or print-quality pdf preferred). The format should be produced directly from original creation package, or original software format.
- 3) Electronic Supplementary Material (ESM): all supplementary materials accompanying an accepted article will be treated as in their final form. Note that the Royal Society will not edit or typeset supplementary material and it will be hosted as provided. Please ensure that the supplementary material includes the paper details where possible (authors, article title, journal name). Supplementary files will be published alongside the paper on the journal website and posted on the online figshare repository (<https://figshare.com>). The heading and legend provided for each supplementary file during the submission process will be used to create the figshare page, so please ensure these are accurate and informative so that your files can be found in searches. Files on figshare will be made available approximately one week before the accompanying article so that the supplementary material can be attributed a unique DOI. Alternatively you may upload a zip folder containing all source files for your manuscript as described above with a PDF as your "Main Document". This should be the full paper as it appears when compiled from the individual files supplied in the zip folder.

Article Funder

Please ensure you fill in the Article Funder question on page 2 to ensure the correct data is collected for FundRef (<http://www.crossref.org/fundref/>).

Media summary

Please ensure you include a short non-technical summary (up to 100 words) of the key findings/importance of your paper. This will be used for to promote your work and marketing purposes (e.g. press releases). The summary should be prepared using the following guidelines:

*Write simple English: this is intended for the general public. Please explain any essential technical terms in a short and simple manner.

*Describe (a) the study (b) its key findings and (c) its implications.

*State why this work is newsworthy, be concise and do not overstate (true 'breakthroughs' are a rarity).

*Ensure that you include valid contact details for the lead author (institutional address, email address, telephone number).

Cover images

We welcome submissions of images for possible use on the cover of Proceedings A. Images should be square in dimension and please ensure that you obtain all relevant copyright permissions before submitting the image to us. If you would like to submit an image for consideration please send your image to proceedingsa@royalsociety.org

Once again, thank you for submitting your manuscript to Proceedings A and I look forward to receiving your revision. If you have any questions at all, please do not hesitate to get in touch.

Best wishes

Alice Power
Publishing Editor
Proceedings A
proceedingsa@royalsociety.org

on behalf of
Professor H.J.S. Fernando
Board Member
Proceedings A

Reviewer(s)' Comments to Author:

Referee: 1

Comments to the Author(s)
Please see attached comments.

Referee: 2

Comments to the Author(s)
See attached pdf file.

Referee: 3

Comments to the Author(s)

The paper is a pleasure to read, insightful and gives a perspective and review of casting decision relevant climate information under (large) uncertainty. A framework is discussed where dominance of epistemic uncertainty leads to an alternative way of obtaining and presenting decision relevant climate information. Storytelling as a means to deal with the subjectivity and complexity of compounded (conditioned or truncated) probabilities is described and a causal network approach is laid out where this is reconciled with traditional frequentist approaches. I find this framework the most interesting part of the paper, but I am biased myself as I worked on storyline approaches.

Given the relevance and importance of the subject and the increasing scientific discussions on this, in particular on the methodology, in the physical climate science community, I would like to see this paper published. The scientific novelty and rigor could be stronger, but as a perspective this is a very relevant and good contribution. It serves the purpose of making physical climate scientists aware how climate information are conveyed and it offers a framework beyond traditional flow of information from global models to impact and decisions. I agree with the author that this traditional approach has very serious problems. The paper is not entirely original though, but it is very nice to see some new aspects addressed. The paper is more of the nature of a perspective paper. I have some reservations which can be addressed by the author:

- One of them is the lack of acknowledgment of existing literature and approaches. The author is likely aware of previous work, but could cite relevant literature more. In particular most elements of the first part the paper on different types of uncertainties have been published in detail before, including the consequences and relevance for decision making for climate (both mitigation and adaptation). The work of Petersen (now at UCL), including his PhD thesis (published as a book as well), goes in great depth on this subject. Also, earlier work of Dessai, Hulme, Lempert and van der Sluijs from the early 2000s onward is scarcely referenced while many of the thoughts put down here can be found in those papers as well. They have been systematically addressed. Again, both for the global mitigation problem as well as the regional climate adaption problems this has been done. The current paper makes the connection to the physical climate science more clear though.

- The author should be more careful with the wording on ethics. I don't think this paper goes into sufficient depth on norms and values to include strong statements on ethics (e.g. the wording in the abstract is rather strong compared to the content of the paper). Although of large relevance, it takes another study to highlight and reason on ethics (and indeed, this has been done as well, but often outside the realm of the physical climate scientists). Also, we should be aware that apparently the traditional approaches are very decision relevant as well, as they have had impact on regional and local adaptation decisions (e.g. spatial planning, infrastructures). To call these approaches almost unethical is debatable.

- The separation of dynamical and thermodynamical processes of climate change can be problematic, as the author acknowledges, as they cannot be separated well. This particular true for tropical and global circulation patterns where convection is important, but for many regional climate adaptation problems in the midlatitudes the separation is mostly useful (but not for summer continental situations). For example, this is clear from the successful Dutch climate scenarios which already separated a regional expression of the global mean temperature change and the regional dynamical changes into 4 possible futures. This was published in 2006 already and in 2014 followed up. In that year these scenarios were complemented by storylines or Tales of current and future weather analogues where the underlying hypothesis was that dynamical changes are harder to predict or project than thermodynamic responses given a dynamical

pattern. Indeed the paradigm approach of Kuhn could already be applied to those developments. The current reflection of dynamics - thermodynamic separation therefore needs more embedding in existing literature and practices.

- A very relevant aspect of the suggested approach is the plausibility of the scenario or storyline. I would like to see more discussion on this. It is only very shortly mentioned at the end of the paper. In particular in relation to the plausibility of dynamical change in global models that drive regional models (with factual, surrogate warming, counterfactual etc boundary conditions).

- Even though this paper is about decision relevance, it takes very much a physical climate scientist point of view only. Most of the discussion and the references to literature is on the physical climate aspect. All the figures are traditional IPCC-like climate variables and metrics and still far away from the practice of decision making in the public or private sector. The land use example is a bit of an exception, but still this is not on decisions, but shows that at regional scales the boundary conditions of the physical climate system conditions the response. I don't see this as a fundamental problem, but it would be good if the author acknowledges that this is the case. There is a wealth of possibilities there, some examples would help.

In summary, I think this is a very useful contribution to the discussion on decision relevant regional climate change information.

Wilco Hazeleger

Author's Response to Decision Letter for (RSPA-2019-0013.R0)

See Appendices A & B.

Decision letter (RSPA-2019-0013.R1)

16-Apr-2019

Dear Professor Shepherd

I am pleased to inform you that your manuscript entitled "Storyline approach to the construction of regional climate-change information" has been accepted in its final form for publication in Proceedings A.

Our Production Office will be in contact with you in due course. You can expect to receive a proof of your article soon. Please contact the office to let us know if you are likely to be away from e-mail in the near future. If you do not notify us and comments are not received within 5 days of sending the proof, we may publish the paper as it stands.

Your article has been estimated as being 19 pages long. Our Production Office will inform you of the exact length at the proof stage.

Proceedings A levies charges for articles which exceed 20 printed pages. (based upon approximately 540 words or 2 figures per page). Articles exceeding this limit will incur page charges of £150 per page or part page, plus VAT (where applicable).

We are keen to promote all published material in the journal. If you wish us to highlight the publication of your paper to appropriate colleagues, please send me by return email the names and email addresses of up to 5 people and we will ensure that they are notified once the paper goes online.

Under the terms of our licence to publish you may post the author generated postprint (ie. your accepted version not the final typeset version) of your manuscript at any time and this can be made freely available. Postprints can be deposited on a personal or institutional website, or a recognised server/repository. Please note however, that the reporting of postprints is subject to a media embargo, and that the status the manuscript should be made clear. Upon publication of the definitive version on the publisher's site, full details and a link should be added.

You can cite the article in advance of publication using its DOI. The DOI will take the form: 10.1098/rspa.XXXX.YYYY, where XXXX and YYYY are the last 8 digits of your manuscript number (eg. if your manuscript number is RSPA-2017-1234 the DOI would be 10.1098/rspa.2017.1234).

For tips on promoting your accepted paper see our blog post:
<https://blogs.royalsociety.org/publishing/promoting-your-latest-paper-and-tracking-your-results/>

On behalf of the Editor of Proceedings A, we look forward to your continued contributions to the Journal.

Sincerely,

Alice Power
Publishing Editor
Proceedings A
proceedingsa@royalsociety.org

Appendix A

N.B. Reviewer comments are in italic font; author responses are in plain font.

Reviewer 1

I find this paper extremely interesting. It provides a highly original, critical but constructive view on the scientific approach required to obtain regional climate-change information. It is shown that ethical decisions are implicit when quantifying and communicating scientific information about regional climate change and the so-called “storyline approach” is proposed as a meaningful way to identify self-consistent and plausible pathways of how particular actions affect regional climate. I very much enjoyed reading this well written and thought-provoking paper and recommend to accept it for publication in RSPA subject to few minor revisions.

Many thanks for these encouraging words!

Minor comments

1) L39: correct spelling of “Fischer”.

Corrected

2) L54: maybe here or in other places: the paper by Pfahl et al. (2017) might be relevant for this distinction of thermodynamic and dynamic effects on extreme events. Pfahl, S., P. A. O’Gorman, and E. M. Fischer, 2017. Understanding the regional pattern of projected future changes in extreme precipitation. Nature Clim. Change, 7, 423–427.

I have added a reference to Pfahl et al. (2017) later in the paper where the distinction between thermodynamic and dynamic effects is discussed.

3) L212: please explain how the multi-model ensemble approach mingles the two sources of uncertainty.

A sentence has been added to this effect, including a reference to Tebaldi and Knutti (2007).

4) L221: “manifested” might be too strong here, I can think of internal variability that is not related to coherent modes such as ENSO.

Changed to “includes”.

5) L311 and in other places: I would prefer past tense (“showed”) because of the specified year of the publication.

This is a stylistic matter, which I am happy to leave to the copy-editor!

6) L333: I am not sure that I can follow this argument. My understanding is that the

storyline approach replaces “What will happen?” by “What can happen?” and it then offers plausible scenarios – but this is not what is written in this paragraph. Is my view too simplified / wrong?

The confusion arises from the fact that the first sentence of the paragraph was a conclusion arising from the arguments within that paragraph, rather than from the previous paragraph (as might have been suggested from its placement). I have moved the sentence to later in the paragraph, which should avoid this problem.

7) About on p. 10 I started to wonder why the aleatoric uncertainty is no longer discussed. I am very much interested in interannual variability and in how it changes in a future climate. And sometimes I get the impression that the climate change community is too much focusing on mean trends and neglecting interannual variability. Even in a hypothetical equilibrium climate, e.g., seasons differ from year to year. I find the “storm track shift” an interesting example: What does it mean that the storm track shifts poleward (assuming it does)? Could it be that in most seasons that storm track moves equatorward and, in a few seasons, it moves poleward and this, on average, is interpreted as a poleward shift? I wondered if this might lead to a second category of storylines, in order to provide plausible and dynamically consistent portrayals of, e.g., weather conditions in a season. Are storylines also ideal to cope with aspects of aleatoric uncertainty, which is highly relevant for interannual variability?

Yes, absolutely. This is the concept that Hazeleger et al. (2015) refer to as “tales of future weather”. It is discussed extensively in Shepherd (2016) and Shepherd et al. (2018). I don’t want to belabour the point here because it is not germane to the main goal of this paper, but I have added some text in Section 5 to make clear that storylines can also be of the internal variability.

8) L458: I don’t fully understand why storylines are “in contrast” to the probabilistic approach. I find this contrast somehow exaggerated, also because of the statement then on L536 – a statement that I expected earlier in the paper.

Storylines are deterministic and do not have probabilities attached. Instead, the emphasis is on plausibility, which is inherently subjective. So there is definitely a disconnect between storylines and the probabilistic approach. I have tried to make this more explicit with additional text where the storyline concept is introduced in Section 4.

9) L470: Notation: is “pa” one variable or a product of p and a? Strange notation.

This is Pearl’s notation. I agree it seems a bit strange to use two letters for one variable, although pa is actually a collection of other x ’s. I suppose Pearl made that choice because to use “ p ” for parent could be confused with p for probability. On balance, I think that using “ P ” and “ p ” for different things in the same equation would be even more confusing. Since Pearl is quoted directly for (2), and I introduce a concrete example later, I prefer to stick with Pearl’s notation here.

10) L475: *I don't see how a storyline can be described as $x_i = x_i'$... x_i is only one variable, but a storyline, I assume, is always multi-dimensional (given by a combination of variables)?*

A storyline could be multi-dimensional (or multivariate) but it doesn't have to be. For example, a particular global warming level could be considered a storyline. And here I am representing dynamical conditions symbolically as a single node, which would of course become multivariate when unpacked. A sentence has been added to say that multivariate storylines can be treated by repeated application of (2).

11) L486: *The figure helps understanding but it would still be helpful to introduce the meaning of H , D , R ... in the text.*

The symbols were actually all defined in the text, but only as they are explicitly discussed. I think it would be tedious to list all the variables up front, given they are defined in the figure, but I have moved the definitions of H , D and R to earlier in the text, which should help.

Reviewer 2

This fascinating paper formulates and discusses the challenge of constructing useful climate information on the regional scale, given the uncertainties that face us, while still retaining the relevant information regarding climate risk.

Storylines have been presented as possible ways to interpret past evidence and project into the future, taking all these factors into account, but they may seem to be too subjective to serve their scientific aims. Shepherd argues persuasively that the usual methods of presenting climate information are not quite as objective or value-free as they would seem to be, and that storylines are not dangerously subjective, as they have been presented as being.

Shepherd presents new rationales for the storyline approach as well as a causal modeling interpretation of it in this piece, arguing persuasively for added care to the distinctions between forms of uncertainty at play in regional climate analyses. I find the paper to need some minor revisions, but well-formulated throughout. I strongly recommend publication of this thought-provoking piece in the Proceedings, once the minor revisions are made. It represents some extremely important advancements in our understanding of the Storyline approach, and as such, deserves publication in the Proceedings. Detailed comments and suggestions follow, below.

Many thanks for these encouraging words!

++++

Line 74: it would be helpful if the author, when introducing the term "aleatoric",

simultaneously introduced the meaning/definition of the term.

Aleatoric has now been defined here (as random).

The introduction of the author's first example of a reliable yet uninformative example from the IPCC of the global water cycle is both informative and compelling.

Thank you. No change required.

What aspects of regional climate change are of concern to the author, being referred to in line 150? It would be most helpful if these were spelled out more explicitly. He hasn't given us an example of what kind of information would be provided in the kind of "informativeness" he would like, so I am uncertain what is being referred to here—is it the details for climate risk, "which could have been articulated"? An example here would have been very illuminating.

The examples of drought and storminess have been added, and a reference to Zappa and Shepherd (2017) has been made after "which could have been articulated", to make this concrete. (Zappa and Shepherd (2017) deal with both drought and storminess in relation to storm track changes.)

The argument that climate science cannot be value free is clear and compelling. The further argument that the global scale itself is normative is an extremely interesting point, argued elsewhere by Shepherd and Sobel (2019), and the fact that the local and regional is always treated as accidental has serious consequences, as they correctly highlight. A few words more about what Fricker's "hermeneutical injustice" refers to would help readers not familiar with her work. Also, if this paragraph could be summarized through saying something like: "because those from the global south could never meet the standards of evidence required to show damage from climate change,...", that would be very helpful.

Unfortunately, the Shepherd and Sobel (2019) paper has gone into journal limbo, so this paragraph has had to be removed.

The picturing of the problem with models illustrated with the bi-modal probability density function on line 298 is particularly apt and helpful.

Thank you. No change required.

By moving the "climate risk problem" from the "prediction space" into the "decision space", outside the domain of what he calls "pure climate science," the author highlights the two elements of the system he wishes to focus on, namely the subjectivity of the epistemic uncertainties in the relevant information, and the reliance of regional climate information on personal or shared values. This move seems entirely appropriate. (But it would be very helpful, here, if the author would define what he meant by "decision space.") By emphasizing the subjectivity of the epistemic

uncertainties, Shepherd rejects the legitimacy of those who wish to claim an “objective” viewpoint for their probabilistic approaches that is, indeed, unjustified. Representing uncertainty instead through a series of many discrete storylines seems to be a giant step forward in the progress towards climate understanding of risk and decision making.

Thank you. I have moved the first mention (apart from the outline in the Introduction) of “decision space” from the first paragraph of Section 4, where indeed there is no definition provided, to the second paragraph, where it becomes clear that it corresponds to posing the question in terms of interventions (or decisions) rather than predictions.

Nice contrast between the “recurrence likelihood of a particularly damaging storm” and “how much worse the flooding would be in a warmer, moister climate”. (Lines 423-424)

Thank you. No change required.

The example of French fields and the 2003 heat wave is incredibly effective at evoking admiration about how much information could be extracted from a single day’s data, vs. a probabilistic or statistical requirement of information. (Lines 430-448) This example truly illustrates the damage done by aggregating information over inhomogeneous populations. The role of storyline accounts in taking advantage of contextual information is extremely useful in this regard.

Thank you. No change required.

It would likely be helpful to those unfamiliar with probability theory to insert somewhere in the Bayesian causal modeling discussion to remind the reader that Bayesian probabilities are subjective or degrees of credence or belief, perhaps on line 536: “by casting storylines within the context of a subjectivist probabilistic framework..”??

Actually there is nothing inherently subjective about a causal network. Even frequentist probabilities can be treated in this way. I have therefore removed the word “Bayesian” from Section 5, to avoid confusion. The subjectivism comes in through the re-framing of the question, which is discussed in Section 6.

The following sentence (lines 598-601) is a particularly strong and crucially important point in this paper. The false impression of objectivity referred to here is likely a source of much resistance to the storyline account itself, I wouldn’t be surprised to discover. Perhaps the author might mention this as a possibility.

“Since epistemic uncertainty is inherently deterministic and subjective, there is no imperative to represent it probabilistically, and probabilistic representations can give a false impression of objectivity.”

Thank you. However I wouldn't want to speculate on why there has been resistance to the storyline approach. That could be counter-productive. I prefer to change people's minds through positive persuasion!

I find the following concluding sentence (lines 662-664), after the clarity and incisiveness of the rest of the paper, to be quite obscure, and not a strength of the paper. Perhaps the author can rewrite or rephrase? "In particular, when it comes to atmospheric circulation aspects of climate change, which are crucial to the treatment of regional climate risk, the primacy of dynamical thinking takes centre stage."

It was intended as a clarion call to atmospheric dynamicists. But I agree it is too parochial. The sentence has been deleted.

Reviewer 3 (Wilco Hazeleger)

The paper is a pleasure to read, insightful and gives a perspective and review of casting decision relevant climate information under (large) uncertainty. A framework is discussed where dominance of epistemic uncertainty leads to an alternative way of obtaining and presenting decision relevant climate information. Storytelling as a means to deal with the subjectivity and complexity of compounded (conditioned or truncated) probabilities is described and a causal network approach is laid out where this is reconciled with traditional frequentist approaches. I find this framework the most interesting part of the paper, but I am biased myself as I worked on storyline approaches.

Many thanks for these encouraging words!

Given the relevance and importance of the subject and the increasing scientific discussions on this, in particular on the methodology, in the physical climate science community, I would like to see this paper published. The scientific novelty and rigor could be stronger, but as a perspective this is a very relevant and good contribution. It serves the purpose of making physical climate scientists aware how climate information are conveyed and it offers a framework beyond traditional flow of information from global models to impact and decisions. I agree with the author that this traditional approach has very serious problems. The paper is not entirely original though, but it is very nice to see some new aspects addressed. The paper is more of the nature of a perspective paper. I have some reservations which can be addressed by the author:

- *One of them is the lack of acknowledgment of existing literature and approaches. The author is likely aware of previous work, but could cite relevant literature more. In particular most elements of the first part the paper on different types of uncertainties have been published in detail before, including the consequences and relevance for decision making for climate (both mitigation and adaptation). The work of Petersen (now at UCL), including his PhD thesis (published as a book as well), goes in great depth on this subject. Also, earlier work of Dessai, Hulme, Lempert and van der Sluijs*

from the early 2000s onward is scarcely referenced while many of the thoughts put down here can be found in those papers as well. They have been systematically addressed. Again, both for the global mitigation problem as well as the regional climate adaption problems this has been done. The current paper makes the connection to the physical climate science more clear though.

The novelty here (as I see it) indeed lies in the applicability of these ideas to physical climate, as well as in the connection between storylines and causal networks (which I have certainly not seen before). I quote from the submitted version:

“The purpose of this paper is to place storylines within a broader epistemological framework.”

“Storylines have much in common with scenario planning and other methods of robust decision-making under uncertainty (Lempert 2013). What is novel is their application to physical climate science, where, perhaps because the system obeys known physical laws, the operative paradigm up to now has been probabilistic, which gives the impression of objectivity.”

So I think I was quite explicit about where the novelty lay. I had also made clear in the submitted version that the issue of epistemic uncertainties in climate models has been pointed out before:

“Therefore, treating epistemic uncertainty as if it were aleatoric, with a focus on the multi-model mean as a best estimate, has no epistemological justification. This has been recognized for some time (Smith 2002; Oppenheimer et al. 2007), but the practice continues to be normative (e.g. as in Figure 1).”

Of course, I could have done more, and I had already noted to myself that I should have referenced the important paper of Dessai & Hulme (2004). But I did explicitly acknowledge how so much of this can be traced back to Funtowicz & Ravetz (1993), which inspired the work of van der Sluijs and colleagues in the Netherlands. And I am relying on the reference to Shepherd et al. (2018), which discusses much of this literature, since it is not the main point of the present paper.

I have now added references to Dessai & Hulme (2004) both for the important distinction between epistemic and aleatoric uncertainty, and for the recognition that probabilities are not always needed. I have also added a reference to van der Sluijs et al. (2005) in support of the statement “the decision framework needs to be one where the subjective and ethical choices are both flexible and transparent”, and to Hazeleger et al. (2015) where the concept of storylines is first introduced.

I have also expanded the references to previous literature in a number of other places, e.g., to Tebaldi & Knutti (2007) for the critique of multi-model ensembles, and to Beven (2011) for the tension between Type 1 and Type 2 errors when considering regional aspects of climate change.

• *The author should be more careful with the wording on ethics. I don't think this paper goes into sufficient depth on norms and values to include strong statements on ethics (e.g. the wording in the abstract is rather strong compared to the content of the paper). Although of large relevance, it takes another study to highlight and reason on ethics (and indeed, this has been done as well, but often outside the realm of the physical climate scientists). Also, we should be aware that apparently the traditional approaches are very decision relevant as well, as they have had impact on regional and local adaptation decisions (e.g. spatial planning, infrastructures). To call these approaches almost unethical is debatable.*

Just as one can use the word “physics” or “chemistry” without being a physicist or a chemist, surely it is acceptable to use the word “ethics” without being a philosopher. It should be clear from the context that I am using “ethics” in its philosophical sense, as involving values. (In this sense, the notion of “unethical” does not even exist.) All I am saying is that the scientific framing of a finding has ethical implications. There is no further judgement being made in such a statement. Nevertheless, I have added a reference to Lloyd and Oreskes (2018) in support of the statement “the way in which climate information is constructed has ethical implications”, which underpins the corresponding statement in the Abstract. The paragraph on hermeneutical injustice went deeper into ethical issues, but has now been deleted (for other reasons).

• *The separation of dynamical and thermodynamical processes of climate change can be problematic, as the author acknowledges, as they cannot be separated well. This particular true for tropical and global circulation patterns where convection is important, but for many regional climate adaptation problems in the midlatitudes the separation is mostly useful (but not for summer continental situations). For example, this is clear from the successful Dutch climate scenarios which already separated a regional expression of the global mean temperature change and the regional dynamical changes into 4 possible futures. This was published in 2006 already and in 2014 followed up. In that year these scenarios were complemented by storylines or Tales of current and future weather analogues where the underlying hypothesis was that dynamical changes are harder to predict or project than thermodynamic responses given a dynamical pattern. Indeed the paradigm approach of Kuhn could already be applied to those developments. The current reflection of dynamics - thermodynamic separation therefore needs more embedding in existing literature and practices.*

I was certainly not claiming that the storyline concept was new in physical climate science; Shepherd et al. (2018) provides many examples of past use. However I do agree that the discussion on the dynamic/thermodynamic separation was a bit truncated, so I have added references to van den Hurk et al. (2014) for the Dutch climate scenarios, and to Pfahl et al. (2017) for a recent application (as suggested by Reviewer 1). It is outside the scope of this paper to delve further into the technical aspects of this separation, which are the subject of active research.

• *A very relevant aspect of the suggested approach is the plausibility of the scenario or storyline. I would like to see more discussion on this. It is only very shortly mentioned at the end of the paper. In particular in relation to the plausibility of dynamical change in global models that drive regional models (with factual, surrogate warming, counterfactual etc boundary conditions).*

As the reviewer knows, the question of how credibly dynamical storylines can be imposed on regional climate models is a technical matter that is of significant current research interest. I would not want to delve into such technicalities here. Rather, the point here is the framing of such an approach. With respect to plausibility, the only point I want to make is that plausibility is an inherently subjective concept, which therefore directly involves the user. No changes made.

• *Even though this paper is about decision relevance, it takes very much a physical climate scientist point of view only. Most of the discussion and the references to literature is on the physical climate aspect. All the figures are traditional IPCC-like climate variables and metrics and still far away from the practice of decision making in the public or private sector. The land use example is a bit of an exception, but still this is not on decisions, but shows that at regional scales the boundary conditions of the physical climate system conditions the response. I don't see this as a fundamental problem, but it would be good if the author acknowledges that this is the case. There is a wealth of possibilities there, some examples would help.*

This would go well outside the author's expertise, and the aim of this paper is to shake up the physical climate science community. Thus I believe it is important to maintain the focus on physical climate science. However, the penultimate paragraph of the submitted version of the paper did make the explicit point that the non-climate aspects of climate risk are often more important than the climate-change aspects. No changes made.

In summary, I think this is a very useful contribution to the discussion on decision relevant regional climate change information.

Thank you.

1 Storyline approach to the construction of regional climate-change 2 information

3 Theodore G. Shepherd, Department of Meteorology, University of Reading, PO
4 Box 243, Earley Gate, Whiteknights, Reading RG6 6BB, U.K.
5 theodore.shepherd@reading.ac.uk, [tel. 0118 378 8957](tel:01183788957)

6 Abstract

[revised manuscript text omitted]

The uncertainty in the climate response to forcing is conceptually very
different. It is not a property of the physical climate system; rather, it is a
property of a state of knowledge, or degree of belief, and it *can* be reduced as
knowledge improves. In contrast to aleatoric uncertainty, which is objective,
such epistemic uncertainty is *subjective* (Kahneman and Tversky 1982).
Therefore, treating epistemic uncertainty as if it were aleatoric, with a focus
on the multi-model mean as a best estimate, has no epistemological
justification. This has been recognized for some time (Smith 2002;
Oppenheimer et al. 2007; Beven 2011), but the practice continues to be
normative (e.g. as in Figure 1). It is interesting to consider why this is so, since
in most areas of science the essential distinction between systematic and
random sources of uncertainty is well recognized. One of the reasons may be
that the extent of the epistemic uncertainty is not particularly well known.
First, climate models are imperfect representations of reality and share many
deficiencies, thus may exhibit a collective bias and fail to explore important
aspects of climate change. Second, even within the world represented by
climate models, the forced circulation response of any particular model is
obscured by internal variability.

As an example of the latter, Deser et al. (2012) estimate that for NH
wintertime midlatitude surface pressure (whose spatial gradient provides an
indicator of circulation changes), ensemble sizes of around 30 are generally
needed to determine the forced decadal changes of a given model over a 45-
306 year period. This is in striking contrast to surface temperature changes,
where the signal-to-noise ratio of the forced response is much larger, and
even single simulations can be informative. One might be tempted to think
that if such a large ensemble size is needed to detect the signal, then the
signal must be small. However, Deser et al. (2012) show that such a change in
surface pressure patterns can alter the risk of regional drought or heavy
precipitation by a factor of two, which is hardly negligible. Most climate
model simulations are performed with much smaller ensemble sizes,
although there is a growing interest in large single-model ensembles in order
to better characterize the epistemic uncertainty within current models.

Another conceptual challenge in dealing with the epistemic uncertainty of
climate change is that the concept of “error” is not well defined. Although in
principle there may be one truth, it is not knowable: there will never be
sufficient observations to define all relevant aspects of future climate; future
climate will in any case be non-stationary; and model projections are based
on climate forcing scenarios that will not be the ones actually realized. Thus,
there has been interest in trying to understand the relationship between
model errors in observable aspects of climate and the forced response
simulated by that model — so-called “emergent constraints” (e.g. Hall and Qu
2006). Such an approach permits a Bayesian probabilistic interpretation of
epistemic uncertainty (Sexton et al. 2012). However, there is a danger that
any such relationship is merely statistical and not causal, and many published

Theodore Shepherd 2019-4-10 9:50 AM

Deleted: might

emergent constraints have been subsequently debunked (see Pithan and
Mauritsen 2013; Simpson and Polvani 2016; Caldwell et al. 2018). In any case,
subjective choices are required in the application of any such constraints.

Theodore Shepherd 2019-4-13 2:28 PM
Deleted: and Simpson

That an aleatoric interpretation of multi-model ensembles can blur the
climate information contained within those ensembles is not difficult to
appreciate. Circulation aspects of climate are related to features such as jet
streams. Over Europe during wintertime, some models show an increase in
jet strength under climate change and others a decrease (see Figure 4 of
Shepherd 2014), moreover the location of the changes varies between models.
Whilst all models predict a significant jet response somewhere, averaging
over the models will lead to a washed-out response. Thus the multi-model
mean may not only be unlikely, but even implausible. The situation is
analogous to the idealized case of a bi-modal Probability Density Function,
whose mean may not be a physically realizable state.

A related issue is apparent in Figure 1. Because precipitation increases in
some regions and decreases in others, the multi-model mean change
inevitably passes through zero, and will be small compared to internal
variability on either side of that line. However, that does not mean that the
change in those regions can be expected to be small compared to internal
variability; it just reflects uncertainty in the sign of the change. When there
are equally plausible futures that point in different directions, averaging
those futures buries relevant information and underestimates risk.

The essential point is that epistemic uncertainties are deterministic, which
means that they introduce correlations; unless those correlations are
accounted for, inferences may be flawed. For example, Madsen et al. (2017)
show that the spread across CMIP5 model projections in temperature and
precipitation changes at the gridpoint scale is significantly exaggerated when
treating the gridpoints independently, as compared to when the models are
ranked by the global mean changes (where the spread comes mainly from
climate sensitivity). This illustrates the general point that with an
inhomogeneous distribution of estimators, one should examine the
distribution of responses to a perturbation rather than the overall response of
the distribution to the perturbation.

4. Re-framing the question

If the construction of regional climate information inevitably involves ethical
choices, then those choices should be made by the users of the climate
information, based on their values. If the uncertainties in the climate
information involve a significant epistemic component, then subjectivity is
inevitable and the epistemic uncertainties similarly need to be
understandable and assessable by the users of the climate information, within
their particular context. Both imperatives move the climate risk problem
outside the domain of pure climate science. Moreover, the recognition that

Theodore Shepherd 2019-4-10 12:40 PM
Deleted: from the prediction space into
the decision space, and

374 epistemic uncertainties are deterministic removes the impulse to provide
probabilities, which can give the illusion of objectivity and thereby reduce
transparency. Instead, epistemic uncertainty can be represented through a
discrete set of (multiple) storylines — physically self-consistent, plausible
pathways, with no probability attached (Shepherd et al. 2018; see also
Hazeleger et al. 2015).

Rather than asking what will happen (as in the traditional, scenario-driven
approach), which we may not be able to answer with any confidence,
storylines allow us to ask what would be the effect of particular interventions
— e.g. different climate forcing scenarios, or different adaptation measures —
across a range of plausible futures. The latter questions are in any case the
societally relevant ones. This re-framing of the climate risk question from the
prediction space to the decision space avoids the confidence straightjacket.
Storylines have much in common with scenario planning and other methods
of robust decision-making under uncertainty (Prudhomme et al. 2010;
Lempert 2013). What is novel is their application to physical climate science,
where, perhaps because the system obeys known physical laws, the operative
paradigm up to now has been probabilistic, which gives the impression of
objectivity.

The different uncertainties that are relevant to climate risk, and the different
human decision points, can be broadly represented as follows. There is
uncertainty in the future climate forcing, which is mainly anthropogenic in
origin, and represents the mitigation options. This combines with the
epistemic uncertainty in climate sensitivity to determine the global-mean
warming level. Whilst there is some aleatoric uncertainty in the global-mean
warming, it is small compared with the forced response on decadal or longer
timescales. A given global-mean warming will be associated with distinct
patterns of regional warming (e.g. land warms more than ocean, the Arctic
warms more than lower latitudes during the winter season), including
changes in lapse rate (Kröner et al. 2017). These regional warming patterns
are largely explainable from thermodynamic principles and thus are fairly
well understood, though have substantial quantitative epistemic uncertainty
(including the possibility of tipping points). A given global-mean warming
level will also be associated with particular dynamical conditions in any
specific region (including the circulation effects of coupled atmosphere-ocean
variability), which have a very large aleatoric component but whose forced
changes are also highly uncertain. The regional warming patterns and
dynamical conditions together produce hazards such as weather or climate
extremes, which then combine with the non-climatic anthropogenic factors of
vulnerability and exposure to create climate impacts.

This representation of the climate risk problem provides a natural framework
for storyline approaches. For example, from the perspective of the Paris
Agreement, one may ask the question of what the climate impacts would be at

Theodore Shepherd 2019-4-5 12:19 PM

Moved down [1]: This re-framing of the climate risk question avoids the confidence straightjacket.

Theodore Shepherd 2019-4-5 12:19 PM

Moved (insertion) [1]

different levels of global-mean warming, and what different mitigation
pathways would lead to those warming levels (IPCC 2018). The epistemic
uncertainty in climate sensitivity now no longer affects the estimation of
climate impacts, but is instead relevant to the carbon budget allowed by the
given level of warming. The epistemic uncertainty in future dynamical
conditions (for a given level of global-mean warming) can then be managed
via storylines, the simplest of which is that the changes in hazard are
dominated by the thermodynamic effects arising from the regional
temperature changes, with **the forced** changes in dynamical conditions
assumed to be negligible. Given the large uncertainties in **the forced**
dynamical changes, this can be considered a reasonable null hypothesis for
climate change (Trenberth et al. 2015; Shepherd 2016), and it is far from
uninformative. It is in fact the basis for all of the predicted changes in
extremes shown in Table SPM.1 of the IPCC AR5 (IPCC 2014a). It also
underlies the “surrogate climate change” (also known as “pseudo-global
warming”) methodology (Schär et al. 1996; Kröner et al. 2017) which is
widely used in regional climate change simulations, and the circulation-
analogue methodology (Cattiaux et al. 2010) which is widely used in extreme-
event attribution. However, specific storylines of forced circulation change
can also be considered (Shepherd 2016; Zappa and Shepherd 2017).

Reframing the climate risk question in this way increases the signal-to-noise
ratio of the climate information by explicitly accounting for the correlated
nature of epistemic uncertainty. An example is provided by Figure 2. The
Mediterranean region receives most of its precipitation during the winter
season, so the predicted wintertime drying of the region, which is a robust
feature of climate model projections (see Figure 1), has important
consequences. The extent of the drying will depend on the global-warming
level, and it is relevant to ask, for instance, what would be the difference
between 1.5C and 2.0C of global warming. However the extent of the drying
will also depend on the pattern of circulation change in the region — an
epistemic uncertainty — which can be characterized by physically coherent
storylines (Zappa and Shepherd 2017). Considering just the range between
the low-impact and high-impact storylines shown in Figure 2, the difference
in drying between 1.5C and 2.0C of global-mean warming under the standard
probabilistic framing is the difference between 0.09 [0.04 to 0.15] and 0.12
[0.05 to 0.20] mm/day (left panel), which would be considered indeterminate
within the stated uncertainties. The storyline framing of the difference is, in
contrast, a deterministic 0.04 vs 0.05 for the low-impact circulation storyline,
0.09 vs 0.12 for the median storyline, and 0.15 vs 0.20 for the high-impact

[revised manuscript text omitted]

Theodore Shepherd 2019-4-12 11:04 AM

Deleted: The advantage of deciding between two alternative hypotheses may also be seen from Bayes’ theorem. The posterior probability of a hypothesis H being true given the data D is (1)

Theodore Shepherd 2019-4-12 11:04 AM

Deleted: $\frac{P(H_2)}{P(H_1)}$

Theodore Shepherd 2019-4-12 11:26 AM

Deleted: .

Theodore Shepherd 2019-4-12 11:04 AM

Deleted: 10

Theodore Shepherd 2019-4-12 11:57 AM

Deleted: In a similar way, moving from the prediction space to the decision space can be seen as a way of regularizing the climate risk problem.

climate change alone but involve many other changing factors, most of which
are highly uncertain. If climate impacts I are a product of hazard H ,
vulnerability V and exposure E , then, conceptually, the anthropogenic changes
in I can be represented as

$$742 \quad \delta I = \delta(HVE) = HV\delta E + HE\delta V + VE\delta H. \quad (10)$$

It may well be that the largest terms on the right-hand side of (10) are the
first two, where it is the combination of climate and weather *variability* with
changing vulnerability and exposure that is the main determinant of climate
risk (Nissan et al. 2019). In this case the decision framework is not so much
that of dealing with climate change as it is that of bringing climate
information into decisions that need to be made in any case. There are calls
for this sort of complex-systems thinking in other areas of science, such as
public health (Rutter et al. 2017): “Instead of asking whether an intervention
works to fix a problem, researchers should aim to identify if and how it
contributes to reshaping a system in favourable ways.”

To return to Kuhn (2012), the construction of regional climate-change
information is not most usefully viewed as a search for an objective truth, but
rather as a search for more complete descriptions of the realities that people
have experienced and may experience in the future, and how those depend on
contingent factors that are under human control. Kuhn’s version of the
Bayesian perspective described above, and the cutting of the Gordian Knot it
enables, is as follows (p. 170): “If we can learn to substitute evolution-from-
what-we-know for evolution-toward-what-we-wish-to-know, a number of
vexing problems may vanish in the process.” In such an enterprise, physical
knowledge of the climate system provides the foundation for the construction
of regional climate information.

Acknowledgements: Useful comments were received from Michaela Hegglin,
Wilco Hazeleger, Lisa Lloyd, and the anonymous reviewers.

Funding statement: This work was supported by the European Research
Council (Advanced Grant 339390).

Theodore Shepherd 2019-4-12 11:04 AM

Deleted: 1

Theodore Shepherd 2019-4-12 11:04 AM

Deleted: 1

Theodore Shepherd 2019-4-10 12:55 PM

Deleted: In particular, when it comes to atmospheric circulation aspects of climate change, which are crucial to the treatment of regional climate risk, the primacy of dynamical thinking takes centre stage.

References

- Adams P, Eitland E, Hewitson B, Vaughan C, Wilby R, Zebiak S. 2015 *Toward*
*an ethical framework for climate services*. A White Paper of the Climate
Services Partnership Working Group on Climate Services Ethics. Available
from www.climate-services.org
- Beven K. 2011 I believe in climate change but how precautionary do we need
to be in planning for the future? *Hydrol. Process.* **25**, 1517–1520.
(doi:10.1002/hyp.7939)
- Beven K. 2016 Facets of uncertainty: epistemic uncertainty, non-stationarity,
likelihood, hypothesis testing, and communication. *Hydrol. Sci. J.* **61**, 1652–
1665. (doi: 10.1080/02626667.2015.1031761)
- Binder J, Koller D, Russell S, Kanazawa K. Adaptive probabilistic networks
with hidden variables. *Machine Learning* **29**, 213–244. (doi:
10.1023/A:1007421730016)
- Bony S, Bellon G, Klocke D, Sherwood S, Fermepin S, Denvil S. 2013 Robust
direct effect of carbon dioxide on tropical circulation and regional
precipitation. *Nature Geosci.* **6**, 447–451. (doi:10.1038/NCEO1799)
- Bukovsky MS, McCrary RR, Seth A, Mearns LO. 2017 A mechanistically
credible, poleward shift in warm-season precipitation projected for the U.S.
Southern Great Plains? *J. Clim.* **30**, 8275–8298. (doi:10.1175/JCLI-D-16-
0316.1)
- Byrne MP, O’Gorman PA. 2015 The response of precipitation minus
evapotranspiration to climate warming: Why the “wet-get-wetter, dry-get-
drier” scaling does not hold over land. *J. Clim.* **28**, 8078–8092.
(doi:10.1175/JCLI-D-15-0369.1)
- Caldwell PM, Zelinka MD, Klein SA. 2018 Evaluating emergent constraints on
equilibrium climate sensitivity. *J. Clim.* **31**, 3921–3942. (doi:10.1175/JCLI-D-
17-0631.1)
- Cattiaux J, Vautard R, Cassou C, Yiou P, Masson-Delmotte V, Codron F. 2010
Winter 2010 in Europe: A cold extreme in a warming climate. *Geophys. Res.*
*Letts.* **37**, L20704. (doi:10.1029/2010GL044613)
- Ceppi P, Zappa G, Shepherd TG, Gregory, JM. 2018 Fast and slow components
of the extratropical atmospheric circulation response to CO₂ forcing. *J. Clim.*
**31**, 1091–1105. (doi:10.1175/JCLI-D-17-0323.1)
- Chadwick R, Boutle I, Martin G. 2013 Spatial patterns of precipitation change
in CMIP5: Why the rich do not get richer in the tropics. *J. Clim.* **27**, 3803–3822.
(doi:10.1175/JCLI-D-12-00543.1)

- | Chandler RE. 2013 Exploiting strength, discounting weakness: combining
| information from multiple climate simulators. *Phil. Trans. R. Soc. A* **371**,
| 20120388. (doi:10.1098/rsta.2012.0388)
- | Coughlan de Perez E, Monasso F, van Aalst M, Suarez P. 2014 Science to
| prevent disasters. *Nature Geosci.* **7**, 78–79. (doi:10.1038/ngeo2081)
- | Deser C, Phillips A, Bourdette V, Teng HY. 2012 Uncertainty in climate change
| projections: the role of internal variability. *Clim. Dyn.* **38**, 527–546.
| (doi:10.1007/s00382-010-0977-x)
- | Deser C, Terray L, Phillips AS. 2016 Forced and internal components of winter
| air temperature trends over North America during the past 50 years:
| Mechanisms and implications. *J. Clim.* **29**, 2237–2258. (doi:10.1175/JCLI-D-
| 15-0304.1)
- | Dessai S, Hulme M. 2004 Does climate adaptation policy need probabilities?
| *Climate Policy* **4**, 107–128. (doi:10.1080/14693062.2004.9685515)
- | Fischer EM, Beyerle U, Knutti R. 2013 Robust spatially aggregated projections
| of climate extremes. *Nature Clim. Change* **3**, 1033–1038.
| (doi:10.1038/NCLIMATE2051)
- | French S, Argyris N. 2018 Decision analysis and political processes. *Decision*
| *Analysis* **15**, 208–222. (doi:10.1287/deca.2018.0374)
- | Funtowicz SO, Ravetz JR. 1993 Science for the post-normal age. *Futures* **25**,
| 739–755. (doi:10.1016/0016-3287(93)90022-L)
- | Gigerenzer G. 2004 Mindless statistics. *J. Socio-Economics* **33**, 587–606.
| (doi:10.1016/j.socec.2004.09.033)
- | Hall A. 2014 Projecting regional change. *Science* **346**, 1461–1462.
| (doi:10.1126/science.aaa0629)
- | Hall A, Qu X. 2006. Using the current seasonal cycle to constrain snow albedo
| feedback in future climate change. *Geophys. Res. Lett.* **33**, L03502.
| (doi:10.1029/2005GL025127)
- | Halpern JY, Pearl J. 2005 Causes and explanations: A structural-model
| approach. Part II: Explanations. *Brit. J. Phil. Sci.* **56**, 889–911.
| (doi:10.1093/bjps/axi148)
- | Hawkins E, Sutton R. 2011 The potential to narrow uncertainty in projections
| of regional precipitation change. *Clim. Dyn.* **37**, 407–418.
| (doi:10.1007/s00382-010-0810-6)

Theodore Shepherd 2019-4-10 12:36 PM

Deleted: Fricker M. 2007 *Epistemic Injustice: Power and the Ethics of Knowing*. Oxford, UK: Oxford University Press.

Theodore Shepherd 2019-4-12 12:00 PM

Deleted: Goodman SN. 2001 Of *p*-values and Bayes: A modest proposal. *Epidemiology* **12**, 295-297. (doi:10.1097/00001648-200105000-00006)

- | Hazeleger W, van den Hurk BJJM, Min E, van Oldenborgh GJ, Petersen AC,
| Stainforth DA, Vasileiadou E, Smith LA. 2015 Tales of future weather. *Nature*
| Clim. Change 5, 107–113. (doi:10.1038/NCLIMATE2450)
- | Held IM, Soden BJ. 2006 Robust responses of the hydrological cycle to global
| warming. *J. Clim.* **19**, 5686–5699. (doi:10.1175/JCLI3990.1)
- | Hoskins B, Woollings T. 2015 Persistent extratropical regimes and climate
| extremes. *Curr. Clim. Change Rep.* **1**, 115–124. (doi:10.1007/s40641-015-
| 0020-8)
- | IPCC. 2014a *Climate Change 2013: The Physical Basis*. Contribution of Working
| Group I to the Fifth Assessment Report of the Intergovernmental Panel on
| Climate Change (Stocker TF, et al., eds.). Cambridge, UK: Cambridge
| University Press.
- | IPCC. 2014b *Climate Change 2014: Impacts, Adaptation, and Vulnerability*.
| Contribution of Working Group II to the Fifth Assessment Report of the
| Intergovernmental Panel on Climate Change (Field CB, et al., eds.). Cambridge,
| UK: Cambridge University Press.
- | IPCC. 2018 *Global warming of 1.5°C*. An IPCC Special Report on the impacts of
| global warming of 1.5°C above pre-industrial levels and related global
| greenhouse gas emission pathways, in the context of strengthening the global
| response to the threat of climate change, sustainable development, and efforts
| to eradicate poverty (Masson-Delmotte V, et al., eds.). Geneva, CH: World
| Meteorological Organization.
- | Kahneman D, Tversky A. 1982 Variants of uncertainty. *Cognition* **11**, 143–157.
| (doi:10.1016/0010-0277(82)90023-3)
- | Kretschmer M, Coumou D, Donges JF, Runge J. 2016 Using causal effect
| networks to analyze different Arctic drivers of midlatitude winter circulation.
| *J. Clim.* **29**, 4069–4081. (doi: 10.1175/JCLI-D-15-0654.1)
- | Kröner N, Kotlarski S, Fischer E, Lüthi D, Zubler E, Schär C. 2017 Separating
| climate change signals into thermodynamic, lapse-rate and circulation effects:
| theory and application to the European summer climate. *Clim. Dyn.* **48**, 3425–
| 3440. (doi:10.1007/s00382-016-3276-3)
- | Kuhn TS. 2012 *The Structure of Scientific Revolutions*, 50th anniversary edition.
| Chicago, USA: The University of Chicago Press.
- | Lauritzen SL, Spiegelhalter DJ. 1988 Local computations with probabilities on
| graphical structures and their application to expert systems. *J. R. Stat. Soc. B*
| 50, 157–224. (http://www.jstor.org/stable/2345762)
- | Lempert R. 2013 Scenarios that illuminate vulnerabilities and robust
| responses. *Climatic Change* **117**, 627–646. (doi:10.1007/s10584-012-0574-6)

- Lloyd EA, Oreskes N. 2018 Climate change attribution: When is it appropriate
to accept new methods? *Earth's Future* **6**, 311–325.
(doi:10.1002/2017EF000665)
- Madsen MS, Langen PL, Boberg F, Christensen JH. 2017 Inflated uncertainty in
multimodel-based regional climate projections. *Geophys. Res. Lett.* **44**, 11,606–
11,613. (doi:10.1002/2017GL075627)
- Maraun D, et al. 2017 Towards process-informed bias correction of climate
change simulations. *Nature Clim. Change* **7**, 764–773.
(doi:10.1038/NCLIMATE3418)
- Ming Y, Ramaswamy V, Chen G. 2011 A model investigation of aerosol-
induced changes in boreal winter extratropical circulation. *J. Clim.* **24**, 6077–
6091. (doi:10.1175/2011JCLI4111.1)
- Mitchell TD. 2003 Pattern scaling: An examination of the accuracy of the
technique for describing future climates. *Climatic Change* **60**, 217–242.
(doi:10.1023/A:1026035305597)
- Nissan H, Goddard L, Coughlan de Perez E, Furlow J, Baethgen W, Thomson
MC, Mason SJ. 2019 On the use and misuse of climate change projections in
international development. *WIREs Clim. Change* **10**, e579.
(doi:10.1002/wcc.579)
- Nuzzo R. 2014 Statistical errors. *Nature* **506**, 150–152.
- O'Neill O. 2002 *A Question of Trust*. The BBC Reith Lectures 2002. Cambridge,
UK: Cambridge University Press.
- Oppenheimer M, O'Neill BC, Webster M, Agrawala S. 2007 The limits of
consensus. *Science* **317**, 1505–1506. (doi:10.1126/science.1144831)
- Pearl J. 2009 *Causality*, 2nd ed. Cambridge, UK: Cambridge University Press.
- Pfahl S, O'Gorman PA, Fischer EM 2017 Understanding the regional pattern of
projected future changes in extreme precipitation. *Nature Clim. Change* **7**,
423–427. (doi:10.1038/nclimate3287)
- Pithan F, Mauritsen T. 2013 Comments on “Current GCMs’ Unrealistic
Negative Feedback in the Arctic”. *J. Clim.* **26**, 7783–7788. (doi:10.1175/JCLI-
D-12-00331.1)
- Prudhomme C, Wilby RL, Crooks S, Kay AL, Reynard NS. 2010 Scenario-
neutral approach to climate change impact studies: application to flood risk. *J.*
Hydrol. **390**, 198–209. (doi:10.1016/j.jhydrol.2010.06.043)

- Rutter H, et al. 2017 The need for a complex systems model of evidence for
public health. *Lancet* **390**, 2602–2604. (doi:10.1016/S0140-6736(17)31267-
9)
- Schär C, Frei C, Lüthi D, Davies HC. 1996 Surrogate climate-change scenarios
for regional climate models. *Geophys. Res. Lett.* **23**, 669–672.
(doi:10.1029/96GL00265)
- Scheff J, Frierson D. 2012 Twenty-first-century multimodel subtropical
precipitation declines are mostly midlatitude shifts. *J. Clim.* **25**, 4330–4347.
(doi:10.1175/JCLI-D-11-00393.1)
- Sexton DMH, Murphy JM, Collins M, Webb MJ. 2012 Multivariate probabilistic
projections using imperfect climate models. Part I: Outline of methodology.
*Clim. Dyn.* **38**, 2513–2542. (doi:10.1007/s00382-011-1208-9)
- Shaw TA, et al. 2016 Storm track processes and the opposing influences of
climate change. *Nature Geosci.* **9**, 656–664. (doi:10.1038/NGEO2783)
- Shepherd TG. 2014 Atmospheric circulation as a source of uncertainty in
climate change projections. *Nature Geosci.* **7**, 703–708.
(doi:10.1038/NGEO2253)
- Shepherd TG. 2016 A common framework for approaches to extreme event
attribution. *Curr. Clim. Change Rep.* **2**, 28–38. (doi:10.1007/s40641-016-
0033-y)
- Shepherd TG, et al. 2018 Storylines: an alternative approach to representing
uncertainty in physical aspects of climate change. *Climatic Change* **151**, 555–
571. (doi:10.1007/s10584-018-2317-9)
- | Simpson IR, Polvani L. 2016 Revisiting the relationship between jet position,
forced response, and annular mode variability in the southern midlatitudes.
*Geophys. Res. Lett.* **43**, 2896–2903. (doi:10.1002/2016GL067989)
- Simpson IR, Seager R, Ting M, Shaw TA. 2016 Causes of change in Northern
Hemisphere winter meridional winds and regional hydroclimate. *Nature Clim.*
*Change* **6**, 65–70. (doi:10.1038/nclimate2783)
- Smith LA. 2002 What might we learn from climate forecasts? *Proc. Natl. Acad.*
*Sci. USA* **99**, 2487–2492. (doi:10.1073/pnas.012580599)
- Tebaldi C, Knutti R. 2007 The use of the multi-model ensemble in
probabilistic climate projections. *Phil. Trans. R. Soc. A* **365**, 2053–2075.
(doi:10.1098/rsta.2007.2076)
- Tebaldi C, Arblaster J. 2014 Pattern scaling: Its strengths and limitations, and
an update on the latest model simulations. *Climatic Change* **122**, 459–471.
(doi:10.1007/s10584-013-1032-9)

Theodore Shepherd 2019-4-10 12:36 PM

Deleted: Shepherd TG, Sobel AH. 2019
Climate and catastrophe: Prediction and
uncertainty. *Public Culture*, under
consideration. .

- Trenberth KE, Fasullo JT, Shepherd TG. 2015 Attribution of climate extreme
events. *Nature Clim. Change* **5**, 725–730. (doi:10.1038/NCLIMATE2657)
- van den Hurk B, et al. 2014 KNMI'14: Climate change scenarios for the 21st
century—a Netherlands perspective. Scientific Report WR2014-01, KNMI, De
Bilt, the Netherlands. (<http://www.climatescenarios.nl/>)
- van der Sluijs JP, Craye M, Funtowicz S, Kloprogge P, Ravetz J, Risbey J. 2005
Combining quantitative and qualitative measures of uncertainty in model-
based environmental assessment: The NUSAP system. *Risk Analysis* **25**, 481–
492. (doi:10.1111/j.1539-6924.2005.00604.x)
- van Niekerk A, Scinocca JF, Shepherd TG. 2017 The modulation of stationary
waves, and their response to climate change, by parameterized orographic
drag. *J. Atmos. Sci.* **74**, 2557–2574. (doi:10.1175/JAS-D-17-0085.1)
- Wears RL. 2003 Still learning how to learn. *Qual. Saf. Health Care* **12**, 471–472.
- Yaniv I, Foster DP. 1995 Graininess of judgment under uncertainty: An
accuracy-informativeness trade-off. *J. Exp. Psych.: Gen.* **124**, 424–432.
(doi:10.1037/0096-3445.124.4.424)
- Zaitchik BF, Macalady AK, Bonneau LR, Smith RB. 2006. Europe's 2003 heat
wave: a satellite view of impacts and land–atmosphere feedbacks. *Int. J. Clim.*
**26**, 743–769. (doi:10.1002/joc.1280)
- Zappa G, Shaffrey LC, Hodges KI, Sansom PG, Stephenson DB. 2013 A
multimodel assessment of future projections of North Atlantic and European
extratropical cyclones in the CMIP5 climate models. *J. Clim.* **26**, 5846–5862.
(doi:10.1175/JCLI-D-12-00573.1)
- Zappa G, Shepherd TG. 2017 Storylines of atmospheric circulation change for
European regional climate impact assessment. *J. Clim.* **30**, 6561–6577.
(doi:10.1175/JCLI-D-16-0807.1)

**Figure 1.** Projected changes in precipitation (in %) over the 21st century
 under a high climate forcing scenario (RCP8.5). Stippling indicates where the
 multi-model mean change is large compared with natural internal variability
 in 20-year means (greater than two standard deviations) and where at least
 90% of models agree on the sign of change. Hatching indicates where the
 multi-model mean change is small compared with internal variability (less
 than one standard deviation), but this does not mean that individual model
 changes are small. From the Summary for Policymakers of IPCC (2014a).

**Figure 2.** Projected average wintertime precipitation change (in mm/day)
 over the Mediterranean basin plotted as a function of global warming level (in
 C) and a 'storyline index' that represents the uncertainty in the pattern of
 circulation change in the region. The high impact storyline corresponds to the
 combination of strong tropical upper tropospheric amplification of surface
 warming and a strengthening of the stratospheric polar vortex, and the low
 impact storyline to weak tropical upper tropospheric amplification of surface
 warming and a weakening of the polar vortex. The light blue dashed line
 represents a magnitude of change that is statistically detectable, and the dark
 blue dashed line to one standard deviation of the interannual variability. In

the left panel, the standard representation of the difference between global
 warming levels of 1.5 C and 2.0 C is shown, taking the low and high impact
 storylines as spanning a range of uncertainty. In the right panel, differences
 are shown conditioned on different storylines. Adapted from Zappa and
 Shepherd (2017).

**Figure 3.** Surface conditions derived from infrared remote sensing for a small
 region in central France, for 1 August 2000 (left panels) and 10 August 2003
 (right panels). The top panels show the normalized difference vegetation
 index (NDVI), with the red colours indicative of vegetation. The lower panels
 show the radiometric temperature, with the colour scale at the bottom. The
 distance scale is shown in the lower-right panel, and the values given in the
 right panels indicate the average differences in those parts of the scene
 between the left and right panels. Adapted from Zaitchik et al. (2006).

**Figure 4.** A causal network describing regional climate risk. The arrows
 indicate the directions of causal influence. See text for details.